# A data and task-constrained mechanistic model of the mouse outer retina shows robustness to contrast variations

**Kyra L. Kadhim**[1,2,*]**, Jonas Beck**[1,2]**, Ziwei Huang**[1,2]**, Jakob H. Macke**[2,3,4]**, Fred Rieke**[5]**, Thomas Euler**[6]**, Michael Deistler**[2,3,7]**, and Philipp Berens**[1,2]

[1]Hertie Institute for AI in Brain Health, University of Tübingen, Tübingen, Germany
[2]Tübingen AI Center, Tübingen, Germany
[3]Machine Learning in Science, University of Tübingen, Tübingen, Germany
[4]Max Planck Institute for Intelligent Systems, Tübingen, Germany
[5]Department of Physiology and Biophysics, University of Washington, Seattle, United States
[6]Institute for Ophthalmic Research, University of Tübingen, Tübingen, Germany
[7]Max Planck Institute for Biological Intelligence, Martinsried, Germany
[*]kyra.kadhim@uni-tuebingen.de

## Abstract

Visual processing starts in the outer retina where photoreceptors transform light into electrochemical signals. These signals are modulated by inhibition from horizontal cells and sent to the inner retina via excitatory bipolar cells. The outer retina is thought to play an important role in contrast invariant coding of visual information, but how the different cell types implement this computation together remains incompletely understood. To understand the role of each cell type, we developed a fully-differentiable biophysical model of a circular patch of mouse outer retina. The model includes 200 cone photoreceptors with a realistic phototransduction cascade and ribbon synapses as well as horizontal and bipolar cells, all with cell-type specific ion channels. Going beyond decades of work constraining biophysical models of neurons only by experimental data, we used a dual approach, constraining some parameters of the model with available measurements and others by a visual task: (1) We fit the parameters of the cone models to whole cell patch-clamp measurements of photocurrents and two-photon glutamate imaging measurements of synaptic release. (2) We then trained the spatiotemporal outer retina model with photoreceptors and the other cell types to perform a visual classification task with varying contrast and luminance levels. We found that our outer retina model could learn to solve the classification task despite contrast and luminance variance in the stimuli. Testing different cell type compositions and connectivity patterns, we found that feedback from horizontal cells did not further improve task performance beyond that of excitatory photoreceptors and bipolar cells. This is surprising given that horizontal cells are positioned to mediate communication across cones and that they add to the model's number of trainable parameters. Finally, we found that our model generalized better to out of distribution contrast levels than a linear classifier. Our work shows how the nonlinearities found in the outer retina can accomplish contrast invariant classification and teases apart the contributions of different cell types.

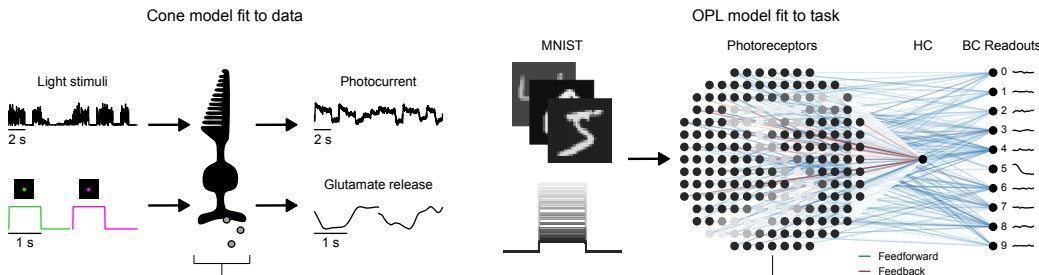

Figure 1: Training a detailed, data-constrained biophysical model of the mouse outer retina to solve a classification task: (left) A cone model with phototransduction cascade and ribbon synapse mechanisms is fit to glutamate and photocurrent recordings. (right) The optimized photoreceptors are spatially distributed and connected to horizontal cells and readout units by excitatory feedforward ribbon synapses. Horizontal cells are also connected to cones by feedback connections. The network is then trained to classify MNIST with varying contrast and luminance.

# 1 Introduction

The outer retina—the first stage of visual processing—already performs nonlinear processing of visual stimuli which shapes downstream vision, and hence, behavior. In particular, these nonlinearities are thought to play an important role in luminance normalization and contrast invariant encoding. Some of the most prominent nonlinearities in the photoreceptors include the phototransduction cascade, which translates the absorption of photons to intracellular currents [1], ion channels which govern the cell's membrane potential [2–5], and the ribbon synapse where glutamate is released dependent on the intracellular calcium concentration [6–8]. In addition, horizontal cells provide recurrent feedback to photoreceptors, further extending the computational capabilities of the outer retina [9].

Previous work has shown that photoreceptors are capable of adapting to drastically different global luminance levels and that this adaptation is visible in the current produced by the phototransduction cascade [10]. It has also been suggested that horizontal cell feedback to photoreceptors facilitates adaptation and contrast enhancement [11]. However, contrast adaptation has been found in bipolar cell dendrites even without input of horizontal cells and without voltage-gated ion channel activity in the bipolar cells [12]. Despite all of this experimental evidence, most computational models of the retina and higher level visual processing either omit the outer plexiform layer or collapse it into spatiotemporal linear filters, relegating nonlinear processing to later stages [13–17].

Understanding the computations of the outer retina requires models that on the one hand incorporate all of the mechanisms previously described, and that on the other hand can be optimized to perform visual processing tasks to constrain parameters that cannot be identified from experimental measurements. However, current biophysically detailed models do not support task-based optimization of parameters [18]. Alternatively, more abstract models of retinal processing, which can be easily optimized for task performance, do not capture biophysical mechanisms at the required level of detail [13, 19–21]. Leveraging newly developed differentiable simulators for biophysical models [22], we built a detailed model of a spatially extended patch of the outer retina including photoreceptors, horizontal cells and bipolar cells, and all of their biophysical mechanisms outlined above (Fig. 1). Using backpropagation of error, we optimized biophysical parameters of the photoreceptors to fit whole cell patch-clamp measurements of photocurrents and two-photon glutamate imaging measurements of synaptic release. We then fit between 2,000 and 2,900 additional parameters of the model not constrained by these experimental measurements to enable the network to perform a classification task. Using this data- and task-constrained model, we then show how the nonlinearities of the outer retina enable a small network of cells with realistic biophysics to perform contrast invariant encoding of visual stimuli and decipher the contributions of different cell types.

# 2 Related work

Few previous works have combined biophysical models with deep learning and optimization techniques [21, 23, 22]. Idrees et al. [21] developed a convolutional neural network with biophysically

realistic photoreceptors as an input layer [21]. This photoreceptor layer was shown to perform adaptation that improved prediction primate and rat retinal ganglion cell activity and enabled better predictions for luminance levels out of the training distribution. Their photoreceptor model, however, did not include ribbon synapses, and had no biophysical or structural realism beyond the photoreceptor layer. They also didn't test for contrast invariant encoding of visual stimuli. Schröder et al. [14] developed a differentiable and biophysically realistic model of the inner retina with realistic synapse dynamics, feedback circuits and trained the model end-to-end, but their work focuses on temporal processing in the inner retina, and they do not constrain their model by task performance.

Other work has used visual tasks to constrain the parameters of more abstract vision models with highly simplified neuron models [24, 23, 25]. For example, Lappalainen et al. [23] recovered experimental properties of neurons in the fly visual system by combining connectome-based constraints and training on an optic flow task, but they used very linear point neurons and simplified synaptic dynamics. We are not aware of any biophysical model of the retina whose parameters were constrained by experimental data and task training, nor of a biophysical model trained to perform a classification task at different contrast levels.

Finally, several biophysically realistic retina models exist [26–28], but they are not implemented in frameworks that are amenable to task optimization, and they often lack biophysical mechanisms. Our model is fully-differentiable, highly-detailed, and implemented in Python, allowing it to interoperate with the Python machine learning ecosystem for efficient gradient-based optimization.

# 3 Methods

## 3.1 A differentiable, biophysically-detailed model of the outer retina

In this section we will provide some background information about our model and its detailed biophysics. In our model, we use single compartment Hodgkin-Huxley type models [29]. These model the flow of different ions through voltage modulated channels in the semi-permeable cell membrane with equations that derive from Ohm's law,

$$I_{\text{ion}}(V, t) = \bar{g}_{\text{ion}} \cdot m^a(V, t) \cdot h^b(V, t) \cdot (V - E_{\text{ion}}), \tag{1}$$

where $V$ is the voltage across the cell membrane, $\bar{g}_{\text{ion}}$ the maximal conductance of a specific ion, $m$ and $h$ the gating variables that model the opening and closing probabilities of the channels, $a$ and $b$ the number of gates or protein subunits per channel, and $E_{\text{ion}}$ is the reversal potential at which there is no net-flow of ions. Given several of these ionic currents and an external input current $I_{\text{input}}$, the membrane potential $V$ is governed by

$$C\frac{\mathrm{d}V}{\mathrm{d}t} = I_{\text{input}} - \sum_{\text{ion}} I_{\text{ion}}, \tag{2}$$

where $C$ is the capacitance. We model different cell-types by including different ionic currents. We model the photoreceptor ion currents after Kamiyama et al. [2] as

$$C\frac{\mathrm{d}V}{\mathrm{d}t} = -(I_{\text{photo}} + I_{\text{Kv}} + I_{\text{hyper}} + I_{\text{Ca}} + I_{\text{K(Ca)}} + I_{\text{Cl(Ca)}} + I_{\text{leak}}). \tag{3}$$

The current generated by the phototransduction cascade $I_{\text{photo}}$ is determined by a complex electrochemical signaling pathway triggered by the absorption of photons by photopigments in the outer segments of photoreceptors. Upon photon absorption, a cascade of intracellular reactions is initiated, ultimately leading to the modulation of ionic currents across the photoreceptor membrane. We model the phototransduction cascade as

$$\frac{\mathrm{d}R}{\mathrm{d}t} = \gamma S(t) - \sigma R(t) \tag{4} \qquad \frac{\mathrm{d}P(t)}{\mathrm{d}t} = R(t) - \phi P(t) + \eta \tag{5}$$

$$\frac{\mathrm{d}G(t)}{\mathrm{d}t} = \frac{S_{\max}}{1 + (C(t)/K_{GC})^m} - P(t)G(t) \tag{6} \qquad S_{\max} = \frac{\eta}{\phi} G_{\text{dark}}\left(1 + \left(\frac{C_{\text{dark}}}{K_{GC}}\right)^m\right) \tag{7}$$

$$I_{\text{photo}}(t) = kG^n(t) \tag{8} \qquad \frac{\mathrm{d}C(t)}{\mathrm{d}t} = \beta\left(\frac{C_{\text{dark}}}{(G_{\text{dark}})^n k} I(t) - C(t)\right). \tag{9}$$

Table 1: Network architectures

| Readout neurons | # Horizontal cells | Connectivity |
|---|---|---|
| Passive | 1 | Full |
| Active bipolar cell | 1 | Full |
| Active bipolar cell | 0 | n/a |
| Active bipolar cell | 9 | Local |
| Active bipolar cell | 200 | One-to-one |

Here, equation 4 models the production and decay of opsin. $S(t)$ is the light stimulus and $R(t)$ the activated opsin molecules. $P(t)$ denotes the second messenger or phosphodiesterase activity; it increases with opsin activation and decays over time (Eq. 5). $G(t)$ corresponds to the concentration of cyclic GMP (cGMP), the key second messenger that controls ion channel opening (Eq. 6). $I_{\text{photo}}(t)$ is the resulting photocurrent, modeled as a nonlinear function of cGMP concentration (Eq. 8). $C(t)$ tracks intracellular calcium concentration, which is modulated by the photocurrent and in turn feeds back to regulate cGMP synthesis (Eq. 9). This feedback system captures the adaptive and nonlinear response dynamics of cone photoreceptors to varying light intensities [10].

Cone photoreceptors also possess specialized ribbon synapses which enable rapid and continuous neurotransmitter release in response to graded intracellular voltage changes [8]. The ribbon synapse facilitates efficient transmission of visual signals to downstream neurons even under high-throughput conditions. It is organized into discrete vesicle pools that undergo cycles of replenishment, priming, and exocytosis. We model the dynamics of the ribbon synapse as

$$\frac{\text{dRP}}{\text{d}t} = d_{\max} \cdot \text{Exo}(t) - r_{\max} \left(1 - \frac{\text{IP}(t)}{\text{IP}_{\max}}\right) \frac{\text{RP}(t)}{\text{RP}_{\max}} \tag{10}$$

$$\frac{\text{dIP}}{\text{d}t} = r_{\max} \left(1 - \frac{\text{IP}(t)}{\text{IP}_{\max}}\right) \frac{\text{RP}(t)}{\text{RP}_{\max}} - i_{\max} \left(1 - \frac{\text{RRP}(t)}{\text{RRP}_{\max}}\right) \frac{\text{IP}(t)}{\text{IP}_{\max}} \tag{11}$$

$$\frac{\text{dRRP}}{\text{d}t} = i_{\max} \left(1 - \frac{\text{RRP}(t)}{\text{RRP}_{\max}}\right) \frac{\text{IP}(t)}{\text{IP}_{\max}} - e_{\max} \cdot f(V(t)) \frac{\text{RRP}(t)}{\text{RRP}_{\max}} \tag{12}$$

$$\frac{\text{dExo}}{\text{d}t} = e_{\max} \cdot f(V(t)) \frac{\text{RRP}(t)}{\text{RRP}_{\max}} - d_{\max} \cdot \text{Exo}(t). \tag{13}$$

In this model $\text{RP}(t)$, $\text{IP}(t)$, and $\text{RRP}(t)$ describe the reserve, intermediate, and readily releasable pool of vesicles, respectively. $\text{Exo}(t)$ indicates the exocytosed vesicles (i.e., neurotransmitter release). Eq. 10 describes the replenishment of the reserve pool from exocytosed vesicles and its depletion into the intermediate pool and the priming of vesicles from the intermediate pool into the readily releasable pool. Eq. 12 & 13 describe the calcium- and voltage-dependent transfer of vesicles from the readily releasable pool into the synaptic cleft and the dynamics of vesicle release and recycling. The term $f(V(t))$ is a sigmoid function of the photoreceptor membrane voltage $V(t)$ capturing the voltage-dependent probability of vesicle release.

The phototransduction cascade and ribbon synapse together allow the transformation of light stimuli into graded voltage signals and their rapid encoding into neurotransmitter output, forming the foundation of early visual processing. With these details, we describe our cone photoreceptors with 26 differential equations, the most detailed computational model of photoreceptors of which we are aware. We implemented all of the mechanisms in an open source Python library.

We model horizontal cells (HCs) based on dynamics of rabbit HCs [3]. These dynamics are are defined by

$$C\frac{\text{d}V}{\text{d}t} = -(I_{\text{K}_{\text{ar}}} + I_{\text{K}_{\text{dr}}} + I_{\text{K}_{\text{to}}} + I_{\text{Na}} + I_{\text{Ca}} + I_{\text{leak}}). \tag{14}$$

For the synapses of cones onto HCs, we use the ribbon presynaptic mechanism described above and an ionotropic post-synaptic mechanism

$$I_{\text{syn}} = \bar{g}_S \cdot \text{Exo}(t) \cdot (V_{\text{post}} - E_{\text{syn}}), \tag{15}$$

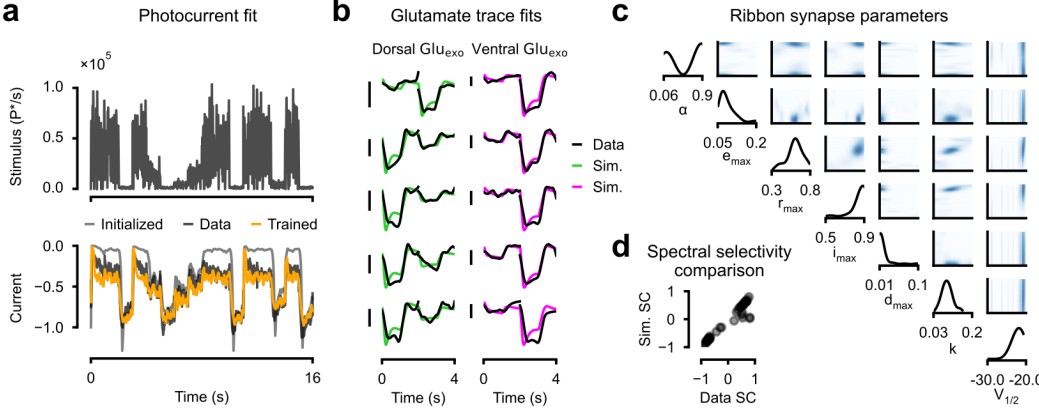

Figure 2: Photoreceptor parameters fit to experimental data using our differentiable simulator. **a)** Phototransduction cascade parameters were fit using gradient descent to normalized photocurrent data when exposed to the light stimulus (top). **b)** Ribbon synapse parameters were fit to glutamate recordings given a green center flash stimulus (2s) and a UV center flash stimulus (2s). Green traces show simulations fit to glutamate recordings from the dorsal retina whereas magenta traces show simulations fit to glutamate recordings from the ventral retina. Scale bars indicate one unit difference from the baseline. **c)** Distributions of fitted ribbon synapse parameters for 42 cones based on one and two dimensional kernel density estimates. **d)** Spectral selectivity, green vs. UV preference, of 42 data traces compared to the trained simulations.

where $g_S$ is the maximal conductance, $\mathrm{Exo}(t)$ is the presynaptic exocytosed glutamate, $V_{\mathrm{post}}$ is the postsynaptic membrane voltage and $E_{\mathrm{syn}}$ is the reversal potential.

For the synapses of HCs onto cones, we use the same ionotropic synaptic mechanism (Eg. 15), but instead of exocytosed glutamate $\mathrm{Exo}(t)$, the synaptic state evolves according to a standard model of ionotropic synapses found in the literature [30]:

$$\frac{\mathrm{d}s(t)}{\mathrm{d}t} = \frac{1}{\tau_s}\left(s_\infty - s(t)\right) \quad s_\infty = 1/\left(1 + \exp\left((V_{\mathrm{th}} - V_{\mathrm{pre}})/\Delta\right)\right) \qquad \tau_s = (1 - s_\infty)/k_-.$$

The exact mechanism of feedback from HCs to cones is still debated [9], so we opted for a phenomenological feedback mechanism.

The bipolar cell channel models were obtained from work observing five main membrane currents in the bipolar cells of several vertebrate species [4]:

$$C\frac{\mathrm{d}V}{\mathrm{d}t} = -(I_{\mathrm{K_A}} + I_{\mathrm{K_v}} + I_{\mathrm{K(Ca)}} + I_{\mathrm{hyper}} + I_{\mathrm{Ca}} + I_{\mathrm{leak}}). \qquad (16)$$

The calcium dynamics were adapted from other work [5]. For the synapses of cones onto bipolar cells, we use the ribbon presynaptic mechanism and the ionotropic post-synaptic mechanism (Eq. 15) such that all bipolar cells are of the OFF subtype, meaning that they hyperpolarize to light-onset.

We evaluated different network configurations for which we varied the number of HCs, connectivity, and bipolar cell dynamics to elucidate which components are important for contrast invariant classification. Table 1 lists the different architectures we compare. Local connectivity means that the HCs are placed on a lattice spanning the diameter of the patch and each is connected to its 50 nearest cones, approximating the cone-HC connection ratio in the mouse retina [31]. The one-to-one connectivity of the 200 horizontal cells is designed to model a case in which each cone photoreceptor gets feedback from an electrically isolated segment of horizontal cell.

We implemented all our models in JAX [32], using JAXLEY [22], a toolbox for differentiable biophysical simulation. This allows our models to run efficiently on GPU and to use gradient-based optimization techniques for model parameter fitting. The code for producing and training our models can be found at `https://github.com/berenslab/jaxley-retina`.

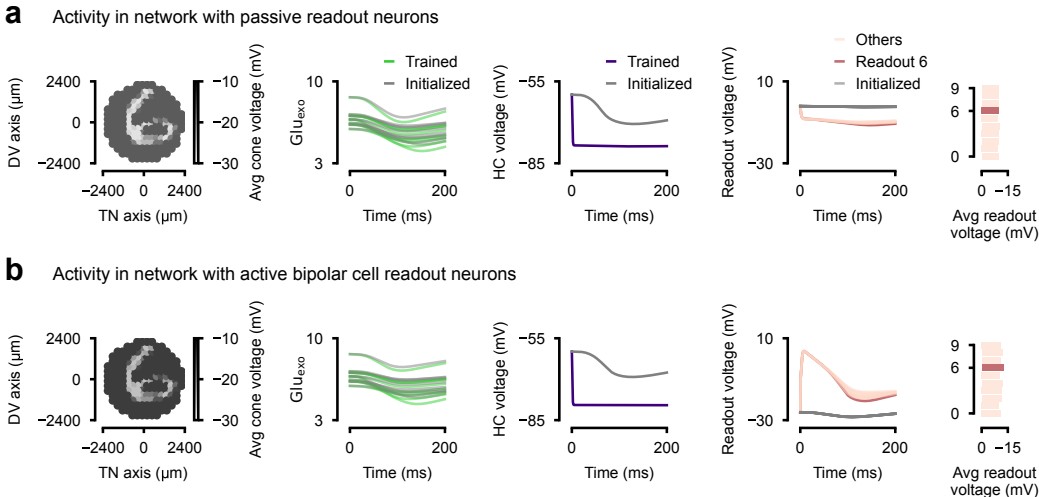

Figure 3: Neural activity in the biophysical model with 200 photoreceptors and one horizontal cell connected to all cones trained to classify MNIST with variance in contrast and luminance. **a)** Activity in the trained architecture with only a passive leak channel on the readout neurons. **b)** Activity in the trained architecture with active bipolar cell channels on the readout neurons.

## 3.2 Data-constraining parameters of the photoreceptor models

The data used for constraining biophysical parameters of the model included recordings of photocurrents from six cones when exposed to 16 seconds of noise stimuli with variable mean [10]. Experimental data was normalized by the experimentally-measured dark photocurrent. To normalize our simulations, we estimated the dark current using model parameters as $kG_{\text{dark}}^n$, where $G_{\text{dark}}$ is the concentration of cGMP in the dark, $n$ is the cGMP channel cooperativity, and $k$ is a constant for converting cGMP concentration to current.

We fit the 11 phototransduction cascade parameters of our cone model by cutting the stimuli and traces into 100 random four second clips, recording the photocurrent state variable in our simulations, and performing gradient descent with a mean squared error (MSE) loss function. All simulations in this work were run with a timestep of 0.025 ms for stable dynamics. We used the Adam optimizer [33] with a learning rate of 0.01 and initialized the parameters following previous work [10]. To ensure that the parameters stayed bounded during optimization and to stabilize training, we used a sigmoid transform, similar to Deistler et al. [22]. The exact bounds can be found in Table S1. We performed leave-one-out cross-validation, leaving the recording from one cone out in each fold to choose a final parameter set for all cones. The entire training required $< 12$ h on a NVIDIA A40.

Ribbon synapse parameters were constrained using fluorescence recordings of the glutamate biosensor iGluSnFR while the cells were exposed to flashes of green and UV light [34]. We used a total of 42 glutamate traces, each with a 1 s green flash followed by a 1 s recovery period and 1 s UV flash followed by a 1 s recovery period. With frozen phototransduction cascade parameters, we recorded the exocytosed glutamate state variable in our simulations and fit six of the parameters in the ribbon synapse model and an additional parameter that scaled the cone's selectivity for green versus UV light ($\alpha$). We again minimized the MSE loss between the simulation and experimentally recorded trace. We used the Adam optimizer with a learning rate of 0.01 and a stopping criteria of <99% the previous average loss over ten epochs. We imposed a maximum number of epochs, 400, and the training for each ribbon synapse typically lasted $\sim$20 hrs on a NVIDIA GeForce RTX 2080 Ti. We also sigmoid transformed the parameters to stay within bounds estimated to maintain simulation stability. The bounds were $(10^{-5}, 1)$ for all rates of vesicle pool movement, $(10^{-5}, 5)$ for $k$, $(-50, -20)$ for $V_{1/2}$, and $(0, 1)$ for the selectivity parameter $\alpha$. Of the 42 parameter sets obtained, we chose the ten best fits of glutamate traces from the ventral retina and the ten best fits of glutamate traces from the dorsal retina for remaining simulations. These were picked as having the lowest MSE while matching the response characteristics of the data (i.e. not only the mean activity).

## 3.3 Task-constraining remaining parameters of the outer retina model

After fitting individual cones to data, we assembled them into a network with varying numbers of horizontal cells and varying readout neuron models. To test the network's ability to encode visual stimuli in a contrast invariant manner, we trained the network to classify MNIST digits with varying contrasts and luminances. Each network had ten bipolar cell readout neurons, one per digit.

For each digit, a contrast and luminance scaling factor was randomly chosen from the range (0.1, 1.0) and (0, 0.1), respectively. We then took the digits with contrast and luminance variation and scaled their intensities to photoisomerization rates up to 40,000 P*/s (photopic) for all digits. We stimulated cones with the photoisomerization rate associated with the pixel at its location for the first 50 ms of the simulation, and this signal then traveled to the readout neurons. The average voltage of the readout neurons over the simulation period was calculated, rectified, and transformed by the softmax function. The cross-entropy loss of the classification was then minimized via gradient descent. We trained the conductance parameter of each ribbon synapse and ionotropic horizontal cell-to-cone synapse in this setup.

For the training, we used the Adam optimizer on an exponential decay schedule with an initial learning rate of 0.01, a decay rate of 0.9 over 1000 transition steps, and gradient clipping by the global norm. We set a maximum of five epochs with early stopping at a 99.5% reduction in average loss over the previous 10 batches. Access to NVIDIA H100 GPUs permitted a batch size of 32 digits. Training for five epochs on the full MNIST training set and evaluating on the test set required two to three days depending on the network architecture. This lengthy training time results from integrating up to 7,652 differential equations for 8,000 timesteps per gradient step.

We applied this training procedure to five different network architectures (Table 1) to study if the addition of bipolar cell channels and the number and structure of horizontal cell connectivity would improve classification performance on the held out MNIST test set. Later, we trained the architectures on the MNIST train split without contrast and luminance distortions and calculated accuracy on the test split at different contrast levels. We trained the network as before, and the evaluation at different contrast levels required 2-6 hours of compute time on the NVIDIA H100 GPU.

# 4 Results

## 4.1 Biophysical cone models can be fit to experimental data using gradient descent

We first fit the parameters of the phototransduction cascade on the recorded photocurrent data, and we then fit the parameters of the ribbon synapse model on the glutamate imaging data using gradient-based optimization (see Sec. 3.2). We found that the photocurrent of the trained simulation fit the photocurrent data qualitatively well with a mean squared error of 0.0092 and fraction of explained variance of 0.86 for the held out test cell current recording (Fig. 2a). Adaptation was visible in the photocurrent data as the reduction in current after the onset of a larger light stimulus. This effect was not visible at initialization—adaptation only emerged after parameter optimization.

Table 2: Performance of different network architectures on classification of all MNIST digits with contrast and luminance distortions. All network architectures have 200 cones and the listed horizontal cell connectivity and readout neuron mechanisms. All network architectures are trained and tested with the same amount of contrast and luminance variation.

| Readout neurons | # Horizontal cells | Connectivity | Test accuracy |
|---|---|---|---|
| Passive | 1 | Full | 0.42 |
| Active bipolar cell | 1 | Full | 0.62 |
| Active bipolar cell | 0 | n/a | 0.63 |
| Active bipolar cell | 9 | Local | 0.64 |
| Active bipolar cell | 200 | 1 to 1 | 0.64 |

The trained ribbon synapse model matched the recorded glutamate traces in both the ventral and dorsal retina (Fig. 2b). Mean squared errors of the best chosen fits ranged from 0.025 to 0.437 and their Pearson correlations from 0.83 to 0.98. The fitted traces typically had the same spectral selectivity (green vs UV sensitivity) as the traces from the data with only slight deviations (Fig. 2d),

$r = 0.96$. Deviations occurred when the traces were noisy or activity was likely influenced by surrounding cells in the experiment. Parameters were overall well constrained by the data and we did not find significant correlations between the fitted ribbon synapse parameters (Fig. 2c).

## 4.2 Outer retina models with and without horizontal cells can perform a visual task despite contrast and luminance variation

We then assembled 200 of these cones into a network model together with horizontal cells and bipolar cells with different connectivity patterns (see Table 1), and we trained this network to classify MNIST digits with contrast and luminance variation. We tested different model architectures with the goal of inferring which mechanism is most important for task performance. The mechanism that made the biggest difference in performance turned out to be the set of biophysically realistic bipolar cell ion channels (Table 2). The addition of horizontal cells, regardless of abundance or connectivity, did not improve classification accuracy substantially beyond that of cone photoreceptors alone. Notably, this is despite the fact that they contributed 400-900 additional trainable parameters.

The addition of bipolar cell channels to the readout neurons changed the dynamics of the readout neurons substantially (Fig. 3) but did not change the dynamics of the cones or horizontal cells noticeably. The readout neurons do not synapse onto any of the other cells, so no change in the activity of the other cells was to be expected. We also investigated the learned conductances of the ribbon synapses from the photoreceptors to each of the bipolar cell readouts. We found that the spatial patterns in the distribution of conductances somewhat resembled the digit that each readout encodes, indicating that a contrast invariant representation of each digit was learned by the model (Fig. 4). The central spot of high conductance weights for readout neuron one likely arose because ones are often rotated differently but consistently occupy the center of the image.

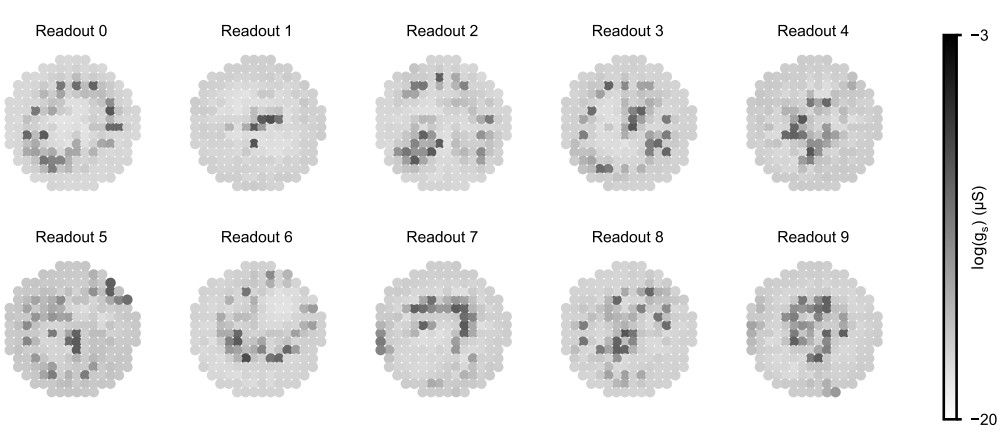

Figure 4: Trained conductances of the photoreceptor ribbon synapses onto each of the readouts in the architecture with 200 photoreceptors and nine locally connected horizontal cells.

## 4.3 Mechanistic outer retina models are robust to out-of-training-distribution contrast changes

For the best performing and most biophysically realistic architecture with 200 photoreceptors and nine locally connected horizontal cells, we tested whether the biophysical model could generalize to contrast levels outside of its training distribution. To this end, we trained the network on the MNIST dataset at full contrast, and we then evaluated the model's performance at lower contrast levels. As references, we compared our model to a linear classifier with 200 units fully connected to ten readout units, a linear classifier preceded by adaptive normalization, a linear classifier preceded by adaptive thresholding, a two-layer multilayer perceptron, and the biophysical model with its initial parameters (Fig. 5a). These models received the same input as the photoreceptors and a softmax was applied to the readout activations to classify the digits.

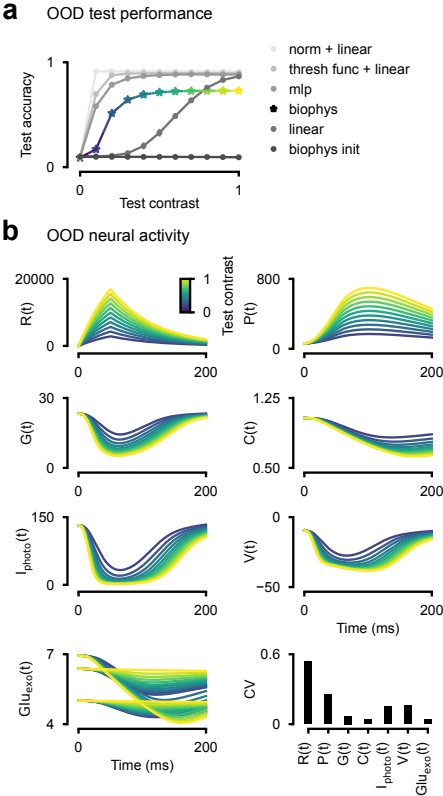

**a** OOD test performance

**b** OOD neural activity

Remarkably, we found that the biophysical model's classification accuracy degraded much more slowly than that of the linear classifier as the contrast level moved further out of the training distribution. However, models that remove contrast variation in the stimuli via normalization or thresholding outperform the biophysical model across contrast levels. Given that the shape of their performance curves are similar, one could suspect that the biophysical model is performing a similar computation, though in a biophysically explicable manner. The MPL outperforms the biophysical model, though it uses far more trainable parameters to do so, and more importantly its computations are not tractable.

For our biophysical model, we are able to explain robustness to out-of-distribution contrast levels with the evolution of state variables. As stimuli is processed, we see large variation in preliminary stages of the phototransduction cascade (activated opsins and phosphodiesterase activity), but then less variation in the following stages where cGMP concentration, calcium concentration, current, voltage, and glutamate release are modulated (Fig. 5b).

## 5 Discussion

Figure 5: Biophysical model outperforms linear classifier on MNIST classifications outside of the contrast training distribution. **a)** Test accuracy at different contrast levels outside of the training distribution (contrast=1.0) of the biophysical model and other reference models. Error bars for the linear classifier and biophysical model across three random seeds are too small to be visible. **b)** Activity of the phototransduction cascade state variables and photoreceptor voltage for a randomly selected cone at the different test contrast levels (first three rows), and glutamate release from three random cones at the different test contrast levels (bottom left). Colors match the contrast levels in **a**. Coefficient of variation across contrast levels for each state variable, averaged over cones (bottom right) with standard error of the means all below 0.02.

Here, we presented a mechanistic model of the mouse outer retina constrained by both experimental data and task performance. Because the model contains a high level of biophysical detail, we were able to analyze how different nonlinear mechanisms found in the outer retina contribute to contrast invariant visual coding. We found architectures with bipolar cell channels on the readout neurons to perform the best, but all network architectures were able to learn the task. Surprisingly, the addition of horizontal cells did not substantially improve classification accuracy in the face of contrast variations or generalization across contrasts, contrary to common believe and despite an increase in the number of trainable parameters. This suggests that horizontal cells, which were experimentally found to modulate cone signals [35, 11, 36], may have a different role in visual processing. While it is possible that different training configurations could have elicited more horizontal cell activation, our study suggests that photoreceptor dynamics, ion channels and ribbon synapses alone are sufficient to transform a light stimulus into a well-scaled, contrast invariant quantity of glutamate release.

Our fitted parameters generally agree with previous estimates for models of the phototransduction cascade (Table S1) and yielded traces that looked similar [10], despite the previous work having trained only a subset of the parameters. Fits of the ribbon synapse parameters could have potentially been improved further with longer training, but we imposed a maximum number of epochs to reduce computational cost. An interesting avenue for future work could be the constraining of photoreceptor parameters with simulation-based-inference as done in previous work [6] and generating simulations that reflect parameter uncertainty.

Previous work has shown that a biophysically realistic photoreceptor model improves CNN predictions of retinal ganglion cell activity and enables the CNN to predict activity given stimuli outside of the network's luminance training distribution [21]. This is a key example of how inductive bias introduced by biophysical models can improve the processing of visual information outside of a model's training distribution. We extend this work and show that a photoreceptor layer with even more biophysical detail and other cell types found in the outer retina can accomplish the classification of images with contrasts it has never seen before.

Limitations of our modeling include the lack of a more detailed horizontal cell to cone synapse. As previously mentioned, the exact mechanisms of this feedback remain unclear. Some hypotheses include various combinations of ephaptic feedback, pH-mediated feedback, and GABA reuptake of horizontal cells [9, 11]. Including a more detailed synapse model would result in a more accurate estimate of horizontal cell activity and its effect on cone dynamics. Such mechanisms could be incorporated in our model in future work and used to study the influence of horizontal cell activity on a variety of computations.

We also did not test the effect of the phototransduction cascade itself on classification at different contrast levels. It has been shown previously that the phototransduction cascade reduces contrast sensitivity due to the introduction of noise depending on the global luminance level [37]. Our model of the phototransduction cascade, however, does not include noise. Future work could study how the addition of noise to the photocurrents affects classification performance at the different contrast levels.

The study of how detailed biophysical models perform computational tasks is not yet commonplace. Biophysical models are often designed and only trained to fit data [26, 38, 14, 21], but then it is not determined what role different pieces of the model play in any computations, such as our analysis of different model architectures. Phenomenological models such as CNNs and linear-nonlinear models are also traditionally fit to data, and some computational insights can be obtained [39, 40], but the underlying biophysics cannot be explained. Our work is a step towards more detailed biophysical models that perform visual tasks. These models can explain computations by analysis of underlying states that map to a wide variety of real physical quantities, as we show for contrast invariant encoding. In this paradigm, more complex tasks, for example including chromatic information and other statistics describing the natural environment, could provide further novel insights into retinal processing.

## Acknowledgements

We thank Sarah Müller and Lisa Schmors for helpful discussions. This work was supported by the Hertie Foundation, German Research Foundation (DFG) through Germany's Excellence Strategy (EXC 2064 – Project number 390727645) and the CRC 1233 "Robust Vision" and the European Union (ERC, "DeepCoMechTome", ref. 101089288, "NextMechMod", ref. 101039115). Views and opinions expressed are however those of the authors only and do not necessarily reflect those of the European Union or the European Research Council Executive Agency. Neither the European Union nor the granting authority can be held responsible for them. KLK, JB, and MD are members of the International Max Planck Research School for Intelligent Systems (IMPRS-IS).

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

# A  Photoreceptor parameters

Table S1: Comparison of fitted photoreceptor parameters to previous estimates

| Parameter | Description | Chen et al. 2024 | Ours | Our bounds |
|---|---|---|---|---|
| $\sigma$ | Opsin decay rate | 9.74 | 14.2 | (5, 24) |
| $\phi$ | PDE decay rate | 9.74 | 12.4 | (5, 24) |
| $\eta$ | PDE dark activation | 761 | 774 | (750, 800) |
| $G_{\text{dark}}$ | Dark cGMP concentration | 15.9-20 | 23.5 | (12, 25) |
| $k$ | cGMP-to-current factor | 0.01 | 0.015 | (0.008, 0.022) |
| $n$ | cGMP channel cooperativity | 3 | 2.88 | (2.8, 3.2) |
| $C_{\text{dark}}$ | Dark $Ca^{2+}$ concentration | 1 | 1.03 | (0.8, 1.2) |
| $\beta$ | $Ca^{2+}$ extrusion rate | 2.64 | 4.42 | (2.5, 10) |
| $m$ | Cooperativity of GC $Ca^{2+}$ dependence | 4 | 4.10 | (3.75, 4.25) |
| $K_{\text{CG}}$ | Affinity of GC $Ca^{2+}$ dependence | 0.4 | 0.348 | (0.2, 0.6) |
| $\gamma$ | Opsin gain | 10 | 10.6 | (1, 22) |

# B  Ion Channel Equations

The implementations of all ion channels can be found in the repository `https://github.com/jaxleyverse/jaxley-mech/tree/main`.

## B.1  Standard form

Gating variables $m$ and $h$ take the following standard form unless otherwise defined.

$$\frac{\mathrm{d}m}{\mathrm{d}t} = \frac{m_\infty(V) - m}{\tau_m(V)} \qquad\qquad \frac{\mathrm{d}h}{\mathrm{d}t} = \frac{h_\infty(V) - h}{\tau_h(V)}$$

$$m_\infty(V) = \frac{\alpha_m(V)}{\alpha_m(V) + \beta_m(V)} \qquad\qquad h_\infty(V) = \frac{\alpha_h(V)}{\alpha_h(V) + \beta_h(V)}$$

$$\tau_m(V) = \frac{1}{\alpha_m(V) + \beta_m(V)} \qquad\qquad \tau_h(V) = \frac{1}{\alpha_h(V) + \beta_h(V)}$$

## B.2  Photoreceptors

Ion channel equations for photoreceptors were found in Kamiyama et al. [2].

$$I_{\text{Kv}} = g_{\text{Kv}} m^3 h (V - e_{\text{K}})$$

$$\alpha_m(V) = \frac{5(100 - V)}{e^{(100-V)/42} - 1} \qquad\qquad \alpha_h(V) = 0.15 e^{-V/22}$$

$$\beta_m(V) = 9 e^{-(V-20)/40} \qquad\qquad \beta_h(V) = 0.4125/(e^{(10-V)/7} + 1)$$

$$g_{\text{Kv}} = 2 \times 10^{-3} \text{ S/cm}^2, e_K = -74 \text{ mV}$$

$$I_{\text{hyper}} = g_{\text{hyper}}(S_1 + S_2 + S_3)(V - e_{\text{hyper}})$$

$$\frac{\mathrm{d}C_1}{\mathrm{d}t} = \beta_h C_2 - 4\alpha_h C_1$$

$$\frac{\mathrm{d}C_2}{\mathrm{d}t} = 4\alpha_h C_1 + 2\beta_h S_1 - (3\alpha_h + \beta_h)C_2$$

$$\alpha_h(V) = \frac{8}{e^{(V+78)/14} + 1}$$

$$\frac{\mathrm{d}S_1}{\mathrm{d}t} = 3\alpha_h C_2 + 3\beta_h S_2 - (2\alpha_h + 2\beta_h)S_1$$

$$\beta_h(V) = \frac{18}{e^{-(V+8)/19} + 1}$$

$$\frac{\mathrm{d}S_2}{\mathrm{d}t} = 2\alpha_h S_1 + 4\beta_h S_3 - (\alpha_h + 3\beta_h)S_2$$

$$\frac{\mathrm{d}S_3}{\mathrm{d}t} = \alpha_h S_2 - 4\beta_h S_3$$

$$g_{\text{hyper}} = 3 \times 10^{-3} \text{ S/cm}^2, \ e_{\text{hyper}} = -32 \text{ mV}$$

$$I_{\text{Ca}} = g_{\text{Ca}} m^4 h (V - e_{\text{Ca}})$$

$$\alpha_m(V) = \frac{3(80 - V)}{e^{(80-V)/25} - 1}$$

$$h(V) = \frac{e^{(40-V)/18}}{1 + e^{(40-V)/18}}$$

$$\beta_m(V) = \frac{10}{e^{(V+38)/7} + 1}$$

$$g_{\text{Ca}} = 0.7 \times 10^{-3} \text{ S/cm}^2$$

Calcium dynamics also involve the calculation of $e_{\text{Ca}}$ and $\text{Ca}_\text{s}$ as follows.

$$e_{\text{Ca}} = -12.5 * \log(\text{Ca}_\text{s}/\text{Ca}_\text{o})$$

$$I_{Ex} = J_{ex} \cdot e^{-(V+14)/70} \cdot \frac{Ca_s - Ca_e}{Ca_s - Ca_e + K_{ex}}$$

$$I_{Ex2} = J_{ex2} \cdot \frac{Ca_s - Ca_e}{Ca_s - Ca_e + K_{ex2}}$$

$$\frac{\mathrm{d}Ca_s}{\mathrm{d}t} = -\frac{10^{-6}(I_{Ca} + I_{Ex} + I_{Ex2})}{2FV_1} - \frac{D_{Ca}S_1(Ca_s - Ca_f)}{\delta V_1} - L_{b1}Ca_s(B_l - Ca_{b,ls})$$

$$+ L_{b2}Ca_{b,ls} - H_{b1}Ca_s(B_h - Ca_{b,hs}) + H_{b2}Ca_{b,hs}$$

$$\frac{\mathrm{d}Ca_f}{\mathrm{d}t} = \frac{D_{Ca}S_1(Ca_s - Ca_f)}{\delta V_2} - L_{b1}Ca_f(B_l - Ca_{b,lf}) + L_{b2}Ca_{b,lf}$$

$$- H_{b1}Ca_f(B_h - Ca_{b,hf}) + H_{b2}Ca_{b,hf}$$

$$\frac{\mathrm{d}Ca_{b,ls}}{\mathrm{d}t} = L_{b1}Ca_s(B_l - Ca_{b,ls}) - L_{b2}Ca_{b,ls}$$

$$\frac{\mathrm{d}Ca_{b,hs}}{\mathrm{d}t} = H_{b1}Ca_s(B_h - Ca_{b,hs}) - H_{b2}Ca_{b,hs}$$

$$\frac{\mathrm{d}Ca_{b,lf}}{\mathrm{d}t} = L_{b1}Ca_f(B_l - Ca_{b,lf}) - L_{b2}Ca_{b,lf}$$

$$\frac{\mathrm{d}Ca_{b,hf}}{\mathrm{d}t} = H_{b1}Ca_f(B_h - Ca_{b,hf}) - H_{b2}Ca_{b,hf}$$

$F = 9.648 \times 10^4$ C/mol, $V_1 = 3.812 \times 10^{-13}$ dm$^3$, $V_2 = 5.236 \times 10^{-13}$ dm$^3$, $D_{Ca} = 6 \times 10^{-8}$ dm$^2$/s, $\delta = 3 \times 10^{-5}$ dm, $S_1 = 3.142 \times 10^{-8}$ dm$^2$, $L_{b1} = 0.4$ s$^{-1}\mu$M$^{-1}$, $L_{b2} = 0.2$ s$^{-1}$, $H_{b1} = 100$ s$^{-1}\mu$M$^{-1}$, $H_{b2} = 90$ s$^{-1}$, $B_l = 500$ $\mu$M, $B_h = 300$ $\mu$M, $J_{ex} = 20$ pA, $J_{ex2} = 20$ pA, $K_{ex} = 2.3$ $\mu$M, $K_{ex2} = 0.5$ $\mu$M, $Ca_e = 0.01$ $\mu$M, $Ca_o = 1600$ $\mu$M.

$$I_{\mathrm{K(Ca)}} = g_{\mathrm{K(Ca)}}m^2n(V - e_{\mathrm{K}})$$

$$\alpha_m(V) = \frac{15(80 - V)}{e^{(80-V)/40} - 1}$$

$$n(\mathrm{Ca_s}, K_{1/2}) = \frac{\mathrm{Ca_s}}{\mathrm{Ca_s} + K_{1/2}}$$

$$\beta_m(V) = 20e^{-V/35}$$

$g_{\mathrm{K(Ca)}} = 5 \times 10^{-3}$ S/cm$^2$, $K_{1/2} = 0.3$ mM, $e_K = -74$ mV.

$$I_{\mathrm{Cl(Ca)}} = g_{\mathrm{Cl(Ca)}}m(V - e_{\mathrm{Cl(Ca)}})$$

$$m(\mathrm{Ca_s}, K_{1/2}) = \frac{1}{1 + e^{(K_{1/2}-\mathrm{Ca_s})/0.09}}$$

$g_{\mathrm{Cl(Ca)}} = 2 \times 10^{-3}$ S/cm$^2$, $K_{1/2} = 0.37$ $\mu$M, $e_{\mathrm{Cl(Ca)}} = -20$ mV.

$$I_{\mathrm{leak}} = g_{\mathrm{leak}}(V - e_{\mathrm{leak}})$$

$g_{\mathrm{leak}} = 0.35 \times 10^{-3}$ S/cm$^2$, $e_{\mathrm{leak}} = -77$ mV.

## B.3 Horizontal Cells

Ion channel equations for horizontal cells were found in Aoyama et al. [3].

$$I_{\mathrm{leak}} = g_{\mathrm{leak}}(V - e_{\mathrm{leak}})$$

$g_{\mathrm{leak}} = 0.5 \times 10^{-3}$ S/cm$^2$, $e_{\mathrm{leak}} = -80$ mV.

$$I_{\mathrm{Na}} = g_{\mathrm{Na}}m^3h(V - e_{\mathrm{Na}})$$

$$\alpha_m(V) = \frac{200(38 - V)}{e^{(38-V)/25} - 1} \qquad \alpha_h(V) = 1000e^{-(V+80)/8}$$

$$\beta_m(V) = 2000e^{-(55+V)/18} \qquad \beta_h(V) = \frac{800}{e^{(80-V)/75} + 1}$$

$g_{\mathrm{Na}} = 2.4 \times 10^{-3}$ S/cm$^2$, $e_{\mathrm{Na}} = 55$ mV.

$$I_{\mathrm{K_{dr}}} = g_{\mathrm{K_{dr}}}m^4h(V - e_{\mathrm{K}})$$

$$\alpha_m(V) = \frac{0.4(65 - V)}{e^{(65-V)/50} - 1} \qquad \alpha_h(V) = \frac{1500}{e^{(V+92)/7} + 1}$$

$$\beta_m(V) = 4.8e^{(45-V)/85} \qquad \beta_h(V) = 0.02 + \frac{80}{e^{(V+100)/15} + 1}$$

$g_{\mathrm{Kdr}} = 4.5 \times 10^{-3}$ S/cm$^2$, $e_{\mathrm{K}} = -80$ mV.

$$I_{\mathrm{K_{to}}} = g_{\mathrm{K_{to}}} m^3 h(V - e_{\mathrm{K}})$$

$$\alpha_m(V) = \frac{2400}{1 + e^{-(V-50)/28}} \qquad\qquad \alpha_h(V) = e^{-V/60}$$

$$\beta_m(V) = 80e^{-V/36} \qquad\qquad \beta_h(V) = \frac{20}{e^{-(V+40)/5} + 1}$$

$$g_{\mathrm{K_{to}}} = 15 \times 10^{-3} \text{ S/cm}^2, \, e_{\mathrm{K}} = -80 \text{ mV.}$$

$$I_{\mathrm{K_{ar}}} = g_{\mathrm{K_{ar}}} m^5(V - e_{\mathrm{K}})$$

$$m(V) = \frac{1}{1 + e^{(V+60)/12}}$$

$$g_{\mathrm{K_{ar}}} = 4.5 \times 10^{-3} \text{ S/cm}^2, \, e_{\mathrm{K}} = -80 \text{ mV.}$$

$$I_{\mathrm{Ca}} = g_{\mathrm{Ca}} m^4(V - e_{\mathrm{Ca}})$$

$$\alpha_m(V) = \frac{240(68 - V)}{e^{(68-V)/21} - 1}$$

$$\beta_m(V) = \frac{800}{e^{(55+V)/55} + 1}$$

$$g_{\mathrm{Ca}} = 3.3 \times 10^{-3} \text{ S/cm}^2, \, e_{\mathrm{Ca}} = 54.176 \text{ mV.}$$

## B.4  Bipolar Cells

Ion channels for the bipolar cell readouts use the leak, $K_V$, KA, and hyperpolarization-activated channels of Usui et al. [4] and the calcium dependent channels of Benison et al. [5]. Calcium dependent channels come from this second source aimed at modeling retinal ganglion cells to reduce the computational complexity of the calcium pump.

$$I_{\mathrm{leak}} = g_{\mathrm{leak}}(V - e_{\mathrm{leak}})$$

$$g_{\mathrm{leak}} = 0.23 \times 10^{-3} \text{ S/cm}^2, \, e_{\mathrm{leak}} = -21 \text{ mV.}$$

$$I_{\mathrm{K_V}} = g_{\mathrm{K_V}} m^3 h(V - e_{\mathrm{K}})$$

$$\alpha_m(V) = \frac{400}{e^{-(V-15)/36} + 1} \qquad\qquad \alpha_h(V) = 0.0003e^{-V/7}$$

$$\beta_m(V) = e^{-V/13} \qquad\qquad \beta_h(V) = 0.02 + \frac{80}{e^{(V+115)/15} + 1}$$

$$g_{\mathrm{K_V}} = 2 \times 10^{-3} \text{ S/cm}^2, \, e_{\mathrm{K}} = -58 \text{ mV.}$$

$$I_{\mathrm{KA}} = g_{\mathrm{KA}} m^3 h(V - e_{\mathrm{K}})$$

$$\alpha_m(V) = \frac{2400}{e^{-(V-50)/28} + 1} \qquad\qquad \alpha_h(V) = 0.045e^{-V/13}$$

$$\beta_m(V) = 12e^{-V/10} \qquad\qquad \beta_h(V) = \frac{75}{e^{-(V+30)/15} + 1}$$

$$g_{\text{KA}} = 35 \times 10^{-3} \text{ S/cm}^2, \; e_{\text{K}} = -58 \text{ mV}.$$

$$I_{\text{hyper}} = g_{\text{hyper}}(S_1 + S_2 + S_3)(V - e_{\text{hyper}})$$

$$\frac{\text{d}C_1}{\text{d}t} = \beta_h C_2 - 4\alpha_h C_1$$

$$\frac{\text{d}C_2}{\text{d}t} = 4\alpha_h C_1 + 2\beta_h S_1 - 3\alpha_h C_2 - \beta_h C_2$$

$$\frac{\text{d}S_1}{\text{d}t} = 3\alpha_h C_2 + 3\beta_h S_2 - 2\alpha_h S_1 - 2\beta_h S_1$$

$$\frac{\text{d}S_2}{\text{d}t} = 2\alpha_h S_1 + 4\beta_h S_3 - \alpha_h S_2 - 3\beta_h S_2$$

$$\frac{\text{d}S_3}{\text{d}t} = \alpha_h S_2 - 4\beta_h S_3$$

$$\alpha_h(V) = \frac{3}{e^{(V+110)/15} + 1}$$

$$\beta_h(V) = \frac{1.5}{e^{-(V+115)/15} + 1}$$

$$g_{\text{hyper}} = 0.975 \times 10^{-3} \text{ S/cm}^2, \; e_{\text{hyper}} = -17.7 \text{ mV}.$$

$$I_{\text{Ca}_{\text{L}}} = g_{\text{Ca}_{\text{L}}} m^2 (V - e_{\text{Ca}})$$

$$\alpha_m(V) = \frac{0.061(V - 3)}{1 - e^{-(V-3)/12.5}}$$

$$\beta_m(V) = 0.058 e^{-(V-10)/15}$$

$$g_{\text{Ca}_{\text{L}}} = 2 \times 10^{-3} \text{ S/cm}^2.$$

$$I_{\text{Ca}_{\text{N}}} = g_{\text{Ca}_{\text{N}}} m^2 h (V - e_{\text{Ca}})$$

$$\alpha_m(V) = \frac{0.1(V - 20)}{1 - e^{-0.1(V-20)}} \qquad\qquad \alpha_h(V) = 0.01 e^{-(V+50)/10}$$

$$\beta_m(V) = 0.4 e^{-(V+25)/18} \qquad\qquad \beta_h(V) = \frac{0.1}{1 + e^{-(V+17)/17}}$$

$$g_{\text{Ca}_{\text{N}}} = 1.5 \times 10^{-3} \text{ S/cm}^2.$$

$$I_{\text{KCa}} = g_{\text{KCa}} \left( \frac{x}{1 + x} \right)(V - e_{\text{K}})$$

$$x = \left( \frac{[\text{Ca}^{2+}]_i}{K_{\text{K}_{\text{Ca}}}} \right)^2$$

$$g_{\text{K}_{\text{Ca}}} = 2 \times 10^{-3} \text{ S/cm}^2, \; K_{\text{K}_{\text{Ca}}} = 0.6 \times 10^{-3} \text{ mM}, \; e_{\text{K}} = -85 \text{ mV}.$$

The calcium pump and equilibrium potential calculation is as follows.

$$i_{Ca} = I_{Ca_N} + I_{Ca_L}$$

$$V_{cell} = \pi r^2 l$$

$$\tau_{eff} = \frac{\tau_{store}}{f_i}$$

$$j_{pump} = v_{pump} \frac{[Ca^{2+}]_i^4}{[Ca^{2+}]_i^4 + K_{pump}^4}$$

$$j_{channel} = \begin{cases} -\dfrac{10000 \cdot i_{Ca}}{2FV_{cell}} & \text{if } -\dfrac{10000 \cdot i_{Ca}}{2FV_{cell}} > 0 \\ 0 & \text{otherwise} \end{cases}$$

$$[Ca^{2+}]_{i,\infty} = [Ca^{2+}]_{eq} + \tau_{store}(j_{channel} - j_{pump})$$

$$\frac{d[Ca^{2+}]_i}{dt} = \frac{[Ca^{2+}]_{i,\infty} - [Ca^{2+}]_i}{\tau_{eff}}$$

$$e_{Ca} = \frac{RT}{2F} \cdot 1000 \cdot \ln\left(\frac{[Ca^{2+}]_o}{[Ca^{2+}]_i}\right)$$

$r = 1$ $\mu$m, $l = 10$ $\mu$m, $F = 96485.3329$ C/mol, $T = 279.45$ K, $R = 8.314$ J/(mol·K), $[Ca^{2+}]_o = 2.0$ mM, $[Ca^{2+}]_{eq} = 1 \times 10^{-4}$ mM, $\tau_{store} = 12.5$ ms, $K_{pump} = 1 \times 10^{-4}$ mM, $v_{pump} = 7.2 \times 10^{-6}$ mM/ms, $f_i = 0.025$.

