# OpenReview forum: "A data and task-constrained mechanistic model of the mouse outer retina shows robustness to contrast variations"
_NeurIPS.cc/2025/Conference — NeurIPS 2025 poster_

### Official Review · Reviewer_x6ob · 2025-06-14

**Clarity:** 4
**Significance:** 3
**Originality:** 3
**Rating:** 5
**Confidence:** 3

**Summary:**

The authors leveraging a biophysically detailed model they fitted outer-retina cone cells using on one hand experimental measurements (patch-clamp and 2photon recordings) and on the other hand a task-driven loss. They implemented their training in a multi-stage process, first fitting the phototransduction cascade using the patch-clamp data, then fitting the synaptic transmission using the 2-photon recordings, and finally optimizing the model to perform a digit classification task with various contrasts.
The model is able to reproduce the experimental data and generalizes well to different contrast levels, showing robustness. Importantly, this contrast robustness is not coming from the horizontal cells, which were thought to be responsible for it.

**Questions:**

- I am not sure about the meaning of the last sentence of section 4.3, could you elaborate further?
- Is the argument of Figure 4 that basically the contrast is filtered out by the photoreceptor cells?

**Ethical Concerns:**

["NO or VERY MINOR ethics concerns only"]

**Final Justification:**

As I wrote from my initial review, I found that this paper is technically solid and is interesting for the field. The author's rebuttal further clarified some points, and I am pleased to keep my initial score.

**Limitations:**

yes

**Paper Formatting Concerns:**

I could not find Tables 3 and 4.

**Quality:**

3

**Strengths And Weaknesses:**

Strengths:
- The multi-stage training process allows for a more structured approach to model fitting, ensuring that each component is accurately represented.
- The model's robustness to contrast variations not being depedent on the horizontal cells seems an interesting and potential important finding.
- The use of both experimental data and task-driven loss provides a comprehensive approach to model fitting, ensuring that the model is not only accurate but also functionally relevant.
- The resulting photoreceptor parameters are consistent with previous studies, which adds credibility to the model.

Weaknesses:
- I found it unfortunate that I could not see Table 3 and 4 that were mentioned in the text, as they would have provided additional insights into the model. On that note, this was the biggest part that I was missing, more detailed ablation studies to understand the contributions of different components of the model to its performance. For example, is a HH model necessary, can we use simpler models for the phototransduction cascade or synaptic transmission?
As we see in the text, running this model is computationally expensive, so it would be interesting to see which model components could be simplified without losing performance.

- I might not sure that the control in Figure 5 is optimal, maybe a two-layer network would be more appropriate.

---

> ### Author Rebuttal · Authors · 2025-07-31
>
> We thank the reviewer for their time in reviewing our manuscript. We appreciate that the reviewer considered our model “accurate but also functionally relevant” and our multi-stage training process “a more structured approach to model fitting.”
>
> To address the reviewer’s first concern about the missing Tables 3 and 4, we apologize that those references were supposed to be to Tables 1 and 2. Table 1 was designed to list the different connectivity patterns and Table 2 was designed to show the performance of each connectivity pattern on the task. We will fix that.
>
> Another concern of the reviewer was that our model is very detailed and thus computationally expensive. We acknowledge this point, and simplifications to the model could be made but at the expense of our scientific understanding of the processing. We are currently able to look at the role of each state variable throughout task performance, and we have created additional plots that provide further insight into the contrast invariant encoding. Specifically, we found that the photopigment and phosphodiesterase activity change greatly as the contrast moves further out of distribution, but that the cGMP concentration is quite robust, and then in turn the downstream calcium concentration, photocurrent and photoreceptor voltage. Such findings would not be possible in a simpler model.
>
> To address the concern about our baseline in Figure 5, we trained and tested some other artificial architectures in addition. We rather refer to these alternative models as “reference models”, as we are not so much interested in their absolute performance, but rather e.g. in their capability to handle out of distribution contrasts. As the reviewer suggested, we tested a two layer MLP with a strong optimization routine but no layer normalization and a linear classifier preceded by layer normalization. Both of these architectures perform better outside of the contrast training distribution than the linear classifier; however, they come with their own set of drawbacks. A two-layer MLP, for example, without layer normalization but with a strong optimization procedure, requires far more trainable parameters by design to achieve higher accuracy. We chose the linear classifier in part because it has a similar number of trainable parameters to the biophysical model, 2,010 vs 2,000. We find that the use of layer normalization in addition to linear classification performs the best, not surprisingly, and the contrast-accuracy curve has a similar shape to that of the biophysical model. Thus, this is likely the computation implemented by our biophysical system, and we can now determine which components are responsible for it.  We are happy to add these reference models to our figure for more context.
>
> To address the reviewer’s questions:
> 1. The last sentence in Section 4.3 describes panel b of Figure 5 where the glutamate release of three photoreceptors is plotted. The glutamate release is plotted for the full contrast stimulus (yellow) for which the model is trained, and then the reduced contrast stimulus at the different colors. These colors match the colors of the line indicating the biophysical model in panel a. This is an illustrative figure showing that as contrast is reduced, some cones release more glutamate than at high contrast whereas others reduce less glutamate than at high contrast. What we should have mentioned, however, is that cones where the digit pixels are present (illuminated cones) tend to increase their glutamate at lower contrasts and cones where the digit pixels are not present (non-illuminated cones in the background) tend to reduce their glutamate release at lower contrasts. This is expected because greater illumination results in hyperpolarization and less glutamate release. These changes in glutamate release maintain the digit representation until contrasts of below 0.3 where the classification accuracy drops as shown in panel a.
> 2. This is closely related to your first question, and hopefully the previous discussion was helpful. The answer is basically yes, the photoreceptors are “filtering out the contrast” in their own way.
>
> We hope that our additional experiments and modifications further increase the confidence of the reviewer in our paper.

---

> > ### Comment · Reviewer_x6ob · 2025-08-04
> >
> > Thank you for your detailed response and I found interesting the interpretation of the function of the layer normalization.

---

### Official Review · Reviewer_N9j5 · 2025-06-23

**Clarity:** 4
**Significance:** 2
**Originality:** 3
**Rating:** 3
**Confidence:** 4

**Summary:**

The paper introduces a fully differentiable biophysical model of the mouse outer retina to understand contrast invariant coding via different cell types. The authors constrained some parameters with measurements while other were optimized on MNIST, but with varying contrast and luminance. The authors show that their model can classify digits in spite of these stimulus variations and identify the most important model mechanism determining classification performance. They also compare their model to a linear classifier and find more robust task performance to out of distribution contrast levels.

**Questions:**

1. Can the authors assess the utility of the biophysical model for contrast invariant processing on this task? Do the complex non-linearities of the biophysical model actually matter for performing the classification task? To me it seems that a simple thresholding operation would be sufficient as a preprocessing step - or at the most normalization plus thresholding. This is the key point for me. I would increase my score if the authors can show that there is more to the model than what simple operations can do in the context of this task.

2. In the out-of-distribution task, how well would the linear model do if trained on a range of different contrast levels? This would provide a better idea of the overall performance that the linear model can reach and help to put the results into context.

3. In the out-of-distribution task, how well would the linear model do if preceded by a simple thresholding operation where the threshold is tuned? This ties into question 1 about how well a simple thresholding operation would help solving the task at different contrast levels but here focusses on the out-of-distribution aspect.

**Ethical Concerns:**

["NO or VERY MINOR ethics concerns only"]

**Final Justification:**

I thank the authors for their detailed responses and engagement in the discussion. My concerns regarding the modified MNIST task still stand. The additional experiments and the promise to frame their findings in a different way alleviate these concerns somewhat but not completely. I still believe that a modified MNIST task was not an appropriate selection, in line with Reviewer RpSH but from a more technical viewpoint. I believe that the required changes to the manuscript are too substantial. I raise my score a little but still tend more towards recommending rejection.

**Limitations:**

yes

**Paper Formatting Concerns:**

No issues.

**Quality:**

1

**Strengths And Weaknesses:**

The paper attempts to address an important problem, which is reconciling biophysical realism and functional task performance. It gets around previous shortcomings of detailed biophysical models by employing differentiable simulators. Using these techniques, the authors construct a detailed model of a patch of the outer retina using single compartment Hodgkin-Huxley type models. They further propose a two phase procedure where first the biophysical parameters are fit to recorded data and then additional parameters are fit to the task. The biophysical model is exceptionally realistic which is a strength of the paper. Using this model the authors tune additional parameters on a variant of MNIST where they change contrast and luminance. Except for the differentiable detailed model, each of these steps uses fairly off-the-shelf methods - bounded gradient descent with MSE loss, and rectified and softmax transformed output voltages with cross-entropy loss. The resulting MNIST test accuracy is not great: between 42% and 64%. Notably, the authors identified bipolar cell ion channels as the most important mechanism for task performance.

The authors also trained their best model without contrast and luminance distortions and tested at different contrast levels, comparing to a baseline linear classifier, which was less robust to contrast variations but achieved higher performance at full and large contrast levels. To me this looks like a very weak baseline and doesn't provide a good reference for out-of-distribution robustness.

Quality:
In my opinion, the task is not well selected and not suitable for supporting the claims about the benefits of the biophysical model for contrast invariant processing. The classification task uses MNIST where contrast and luminance are varied. However, MNIST has no resemblance to a real task that the biological system might face and or would be optimized for. For MNIST, the gray values of pixels do not matter that much: binarized image pixels suffice to reach relatively good performance. Checking just now and using a well trained model, I obtained 67.8% test set accuracy after binarizing the MNIST test set pixels without retraining the model and without optimizing the threshold. This is a better accuracy than what the best of the proposed biophysical models achieves. Simple operations might be all that is needed to achieve the reported MNIST performance. It is not clear whether the task is suitable for assessing contrast invariant processing of the biophysical model.

Clarity:
The paper is very well written. Introduction, Related Work, Methods and Results build on each other and provide context and details. The figures are informative and complement the text but do not always use the available space well (generally lots of whitespace, especially for Fig. 5). I did not try but I do believe that the paper provides enough information to reproduce its results.

Significance:
The paper addresses a general shortcoming of previous work: biophysically detailed models do not support task-based optimization of parameters. Overcoming this problem could open up new possibilities for modelling and understanding biophysical processes. It is not clear how well the proposed approach extends to more complex and more useful tasks and how important is compatibility of the task and the biophysical model. Even the MNIST performance of the proposed models is relatively poor.

Originality:
The authors claim that their biophysical model is the most detailed computational model of photoreceptors. I am not aware of an even more detailed model, so I take that at face value. However, the main contribution of the work is the training procedure where part of the model is constrained by recordings while other parameters are constrained by a task. The related work section does a good job of listing other approaches. From what I can tell, the proposed procedure is novel but might be of limited utility for reasons mentioned above.

Minor:

Eq. 1: Superscript M not explained.

Eq. 3: The ion h can be confused with the closing variable h of Eq. 1.

Eq. 9: Double full stop.

Figure 3 caption: "with only with".

---

> ### Author Rebuttal · Authors · 2025-07-30
>
> We thank the reviewer for their detailed and valuable feedback. The reviewer found our manuscript to be “well written” and providing sufficient details for reproducing our findings. The reviewer also positively noted that we were able to “get around previous shortcomings of detailed biophysical models” and that “overcoming this problem could open up new possibilities for modelling and understanding biophysical processes.” We further appreciate that our model was called “exceptionally realistic” and that our proposed procedure for fitting the parameters was considered novel. Given this positive appraisal of our work, we were surprised by the low score.
>
> The reviewer’s main critique was that our biophysical model did not achieve classification accuracy on par with artificial neural networks and that the biophysical model is more complex than it needs to be to solve the task. We would like to clarify that the main goal of our work is not to compete with ANNs, but to investigate how a biophysically realistic model of the outer retina could implement contrast invariant visual processing and what biophysical mechanisms are at play.
>
> Biophysical systems are much more constrained than artificial architectures because information is transmitted by a variety of bounded physical quantities. We also used experimental data to fit most model parameters which further constrained the model. In this biophysical system, “simple operations” employed by artificial architectures must be implemented as the change of physical quantities, and this is not trivial. We are happy to make this perspective and motivation more clear in the introduction and discussion of our paper.
>
> To provide more biological insights, we also plotted all of the state variables in the phototransduction cascade throughout the classification of out of distribution stimuli at different contrast levels. This figure shows that the photopigment and phosphodiesterase activity change greatly as the contrast moves further out of distribution, but that the cGMP concentration is quite robust, and then in turn the downstream calcium concentration, photocurrent and photoreceptor voltage. Our Fig. 5 currently only shows the glutamate traces as contrast moves further out of distribution, but we are happy to show additional state variables that make predictions about the biophysical mechanisms supporting contrast invariant encoding in even more detail.
>
> The reviewer also cited a lack of strong baselines.  While we consider the linear model as a point of reference rather than as a baseline for our model to beat in terms of absolute classification accuracy, we agree with the reviewer that additional reference models would help to contextualize the performance of the biophysical model. We thank the reviewer for this suggestion and have run additional experiments that we compare to in the revised version of the manuscript.  The additional non-biophysical reference models we explored for this rebuttal include a two-layer MLP, layer normalization followed by linear classification, and adaptive and static thresholding followed by linear classification.
>
> Interestingly, a two-layer MLP achieves much better classification performance for all contrasts than our model, but it uses far more trainable parameters by design to achieve this. Furthermore, the computation this network implements is quite opaque, and the mechanism by which it achieves this feat is unclear.
>
> We find that the use of layer normalization in addition to linear classification performs the best, with a contrast-accuracy curve similar to that of our model. While it provides an explicit description of the computation, it does not provide insights into how such a computation could be implemented by the outer retina, as our model does.
>
> Using static thresholding plus linear classification does not work for classifying out of distribution stimuli in this case because the models are trained at full contrast, and thus the threshold will be too high or too low for all of the other contrast levels depending on where it is initialized. That is, there is no proper way to learn a static threshold that classifies the stimuli well outside of the training distribution. An adaptive threshold, however, that is multiplied by the maximum stimulus intensity, performs almost the same as layer normalization with linear classification and thus implements a similar computation, but likewise does not provide insights into the biophysical implementation of such a computation.
>
> Thus, in summary, the rationale of our revised paper is:
> 1. We hope that our previous discussion about the biophysical realism of our system convinces the reviewer that seemingly simple computational operations are not actually so simple in constrained biophysical systems and our model helps to solve that question. In addition, it is important to note that our model considers time, and this is critical to our testing of the hypothesis that horizontal cells contribute to task performance, because their feedback would be required on an adequate timescale. While this makes our model suitable for answering a host of other interesting questions, in the context of this task, assessing the role of the different cell types is not possible without the notion of time.
>
> 2. As a reference model, we used a linear classifier to see if our biophysical model simply implements linear functions. We found that the linear model performs very well in its training distribution (87% when trained at varying contrasts), but its performance degrades much more quickly as contrast moves outside of the training distribution than for our biophysical model. We thus concluded that the biophysical mechanisms implemented in our model underlie this ability to generalize.
>
> 3. Prompted by the reviewer, we used other reference models and found that the linear model does perform much better when preceded by a simple adaptive thresholding operation on out-of-distribution stimuli, so we suppose that this is the sort of operation that the biophysical model implements, though in a biophysically plausible way with real physical quantities. We are happy to include this in our figure and discussion in the paper.
>
> We are grateful to the reviewer for their feedback and hope that they will indeed raise their score as mentioned in the review in light of these clarifications and forthcoming additions to the paper.

---

> > ### Comment · Reviewer_N9j5 · 2025-08-02
> >
> > I thank the authors for their detailed answers and clarifications, and for the additional experiments involving other reference models. However, I still have concerns regarding the general suitability of the selected task.
> >
> > > [...] Given this positive appraisal of our work, we were surprised by the low score.
> >
> > While there are many positive paper aspects, my score is linked to my main concern regarding the question of whether the task is suitable for investigating contrast invariant processing.
> >
> > > The reviewer’s main critique was that our biophysical model did not achieve classification accuracy on par with artificial neural networks and that the biophysical model is more complex than it needs to be to solve the task. We would like to clarify that the main goal of our work is not to compete with ANNs, but to investigate how a biophysically realistic model of the outer retina could implement contrast invariant visual processing and what biophysical mechanisms are at play.
> >
> > No, my main critique is not about performance relative to an artificial neural network. My concern is about whether the modified MNIST task is suitable for assessing contrast invariant processing.
> >
> > As mentioned in my original review, binarizing MNIST pixels using a simple non-optimized and non-adaptive threshold, and then classifying the resulting images using an artificial neural network without retraining achieves better performance than the best performance reported by the authors. Binarizing in this way discards all contrast and luminance variations. To me this suggests that the _task_ i.e. MNIST classification itself, does not require contrast processing beyond an non-optimized non-adaptive threshold in order to achieve performance similar to what was reported by the authors. This invalidates the _task_ for the experimental question that the authors seek to answer and is not about the artificial neural network. The ANN merely shows that there is enough information carried by the binarized pixels to solve the task at the reported performance level.
> >
> > I hope this clarifies my main concern.

---

> > > ### Author Response · Authors · 2025-08-02
> > > **Task suitability concern**
> > >
> > > Dear Reviewer, thank you for the clarification. The first task in the paper is the classification of digits with contrast and luminance variation where we report test accuracies in Table 2. We understand now that you question the validity of this task. For this task, a static threshold to binarize all pixels, chosen small enough, might actually be sufficient in principle. Importantly, this would be a way of generating a contrast invariant representation of the digits, but the outer retina needs to come up with such a contrast invariant representation of the digits with its available biophysical mechanisms. In addition, in the second task, it is clear that a static threshold is not sufficient to generate a contrast invariant digit representation -  if the threshold is set on the high contrast condition, the performance very quickly deteriorates to chance, while the outer retina model still performs well. An adaptive threshold, however, can solve the task similarly well to the biophysical model. Therefore, we think that our model can be used to study how the outer retina computes an adaptive threshold operation, even outside the regime it has been task trained on. We hope this clarification alleviates your main concern.

---

> > > > ### Comment · Reviewer_N9j5 · 2025-08-02
> > > >
> > > > Thank you for the additional clarifications.
> > > >
> > > > > For this task, a static threshold to binarize all pixels, chosen small enough, might actually be sufficient in principle.
> > > >
> > > > It is sufficient to achieve the reported performance.
> > > >
> > > > > Importantly, this would be a way of generating a contrast invariant representation of the digits, but the outer retina needs to come up with such a contrast invariant representation of the digits with its available biophysical mechanisms.
> > > >
> > > > The outer retina would not necessarily have to come up with such a representation. Using the binarized example, I meant to illustrate that MNIST is ill-suited for studying contrast invariant processing, because this classification task is not inherently contrast dependent. Gray values in the original digit images do not carry all that much information about the digits to begin with.
> > > >
> > > > > In addition, in the second task, it is clear that a static threshold is not sufficient to generate a contrast invariant digit representation.
> > > >
> > > > Sorry, which task are you referring to here? Is it the OOD task? Could you explain in more detail why the situation is fundamentally different here?

---

> > > > > ### Author Response · Authors · 2025-08-03
> > > > >
> > > > > We are happy to provide the additional clarifications, and hopefully the following explanations further help.
> > > > >
> > > > > The first task - the classification task with contrast and luminance variation - is designed so that the model must classify the digits at all contrast levels, and, in our view, this is sufficient to claim a contrast-invariant representation needs to be achieved. To note, it is more difficult to classify digits at lower contrast levels because background and digit pixels stimulate cells more similarly and the differential stimulation needs to be translated into neural signals that allow for downstream processing.
> > > > >
> > > > > The second task - the out of distribution task - is fundamentally different because it demonstrates that the model can generalize outside of its training distribution, which is an important thing for biological systems to do. Also to note here, a static non-linearity/thresholding operation learned on the highest contrast will achieve much lower performance on lower contrasts, because it also clips away the signal at lower contrasts.

---

> > > > > > ### Comment · Reviewer_N9j5 · 2025-08-04
> > > > > >
> > > > > > But would a low static threshold not also solve the second task? And also work on the trainjng set, even though it is not the only parameter to work on the training set?
> > > > > >
> > > > > > Would a task where gray values carry meaning for solving the task not be a better choice for assessing contrast invariant processing?

---

> > > > > > > ### Author Response · Authors · 2025-08-04
> > > > > > >
> > > > > > > Sorry for taking a bit to respond. We performed additional experiments regarding the second task and found that the precise threshold on the high contrast condition did not greatly influence test accuracy (in this condition) and quite low thresholds worked almost equally well. For lower test contrasts, classification performance is maintained so long as the threshold is close to the average pixel luminance, which is held approximately constant across contrast levels. In the biophysical model, classification accuracy is maintained until a contrast factor of about 0.3, comparable to a static threshold of slightly larger than the optimal threshold of average pixel luminance. For us, the question remains: What are the biophysical mechanisms by which the outer retina achieves this computation? Our model helps to answer this question, and without the combination of biophysical modeling and task training, this would be impossible. Based on this helpful discussion with the reviewer, we will extend the discussion of the limitations of this task and appropriately rephrase the framing of our findings.

---

### Official Review · Reviewer_ApBk · 2025-06-25

**Clarity:** 3
**Significance:** 3
**Originality:** 3
**Rating:** 5
**Confidence:** 4

**Summary:**

The manuscript presents a fully differentiable, biophysically detailed model of a mouse outer retinal patch—including cone photoreceptors, horizontal cells, and bipolar cells—implemented in JAX. Cone parameters are constrained by fitting to patch-clamp photocurrent and two-photon glutamate imaging data, after which the assembled network is trained end-to-end on a contrast- and luminance-varying MNIST classification task. The findings are interesting. Firstly, cone phototransduction and ribbon-synapse nonlinearities alone produce contrast-invariant representations sufficient for classification. Secondly, adding horizontal-cell feedback does not measurably improve performance beyond the cones plus bipolars model, contrary to prevailing hypotheses. Finally, the biophysical model generalizes better to out-of-distribution contrasts than a linear baseline.

**Questions:**

1. In Figure 1, on the left panel, I'm unclear what the y-axes for "Photocurrent" and "Glutamate release" actually measure. What do the green and magenta square signals represent? On the right panel, what do the different luminance levels of the MNIST squares correspond to, and how are those mapped to photoreceptor grayscale intensities? Why is only a single cell shown under the HC? The BC readout traces need clear axis labels- what do their x- and y-axes denote? Finally, I don't see Figure 1 referenced in the main text.
2. The paper does not spell out the voltage- and time-dependent formulas for the gating variables $𝑚$ and $ℎ$ in Eq. (1).
3. The authors defer the photoreceptor ion-current formulas to Kamiyama et al. (2024), providing the explicit voltage‐ and time‐dependent expressions for each current term (e.g. $ I_{Kv} $, $ I_{h} $, $ I_{Ca} $, $ I_{K(Ca)} $, $ I_{Cl(Ca)} $, $ I_{Leak} $) and for the gating variables $ m(V, t) $ and $ h(V, t) $ in Eq. (1) would greatly improve clarity. Writing those equations out explicitly (or at least in a Supplementary Material) would also help the readers to understand the model better.
4. In Table 1, the switch from "local" to "one-to-one" connectivity (full connection) seems very abrupt. How does horizontal cell connectivity gradually evolve from local to full connection?
5. In Eq. 9, it has two periods at the end of the equation.
6. Some expressions are not consistent. For example, line 101 "equation", while other places use "Eq." in line 118.
7. I suggest that the authors put the main formulas in the main text and put the detailed derivations in the Supplementary Material.
8. The acronym should be written in the first time it appears in the text. For example, HCs refers to horizontal cells in line 127.
9. Can authors explain that why horizontal cells can perform a visual task despite contrast and luminance variation intuitively? Moreover, I suggest that the authors should add to the abstract to explain the mechanism of this finding.

**Ethical Concerns:**

["NO or VERY MINOR ethics concerns only"]

**Final Justification:**

The authors have thoughtfully addressed my concerns in their rebuttal materials, providing thorough and compelling responses. I recommend this paper for acceptance.

**Limitations:**

1. MNIST offers a highly simplified and low-dimensional dataset, while natural scenes exhibit far richer spatial and color statistics, so it would strengthen the work to evaluate the model on more complex benchmarks (e.g., CIFAR-100, ImageNet, etc.). These tests would demonstrate whether the retina's contrast-gain and invariance mechanisms generalize beyond handwritten digits to real-world structures.
2. Section 2 mentions prior models; thus, the discussion needs to discuss them in detail, especially explicitly comparing your findings with those frameworks.

**Paper Formatting Concerns:**

No major formatting issues in the paper.

**Quality:**

3

**Strengths And Weaknesses:**

**strengths:** The manuscript has novel integration of data and task constraints. Fitting 2,000 and 2,900 biophysical parameters by both experimental recordings and a high-level visual task is a technically impressive advance. Moreover, the photoreceptor model comprises 26 differential equations capturing the transduction cascade, calcium feedback, and ribbon-synapse dynamics. The finding that horizontal-cell feedback adds little to contrast-invariance challenges established theories and opens new questions about horizontal-cell function. Finally, the manuscript begins with code, which is suitable for reproducibility and community extension.

**Weaknesses:** 1.  The cascade model lacks stochasticity, which is important for the retina to perform contrast adaptation.

Reference: Cottaris, N. P., Wandell, B. A., Rieke, F., & Brainard, D. H. (2020). A computational observer model of spatial contrast sensitivity: effects of photocurrent encoding, fixational eye movements, and inference engine. Journal of Vision, 20(7):17. https://doi.org/10.1167/jov.20.7.17

2. In line 136, $ \tau_{s} = (1 - s_{\infty})(k^{-})$ is not correct. Please check the equation in the cited paper (ref. 30, page 20, eq. 18). Please also check if the findings still hold for the correct equation (**This is my major concern**).

---

> ### Author Rebuttal · Authors · 2025-07-30
>
> We thank the reviewer for the detailed assessment of our work and for their valuable feedback. They found our work to be “novel”, inviting to “community extension” and a “technically impressive advance” in the field.
>
> The reviewer raises an interesting point about stochasticity as discussed by Cottaris et al. 2020.  We did include a citation to this paper in our discussion section and discussed how this was a limitation of our current model. However, we would like to reiterate that our model of photoreceptor activity is already highly detailed and the photocurrents are trained on actual data with a noisy stimulus. We also note that most biophysical models do not consider noise, likely because the mathematical form of realistic noise in these systems is difficult to define. This is an interesting problem for future work.
>
> The second major concern of the reviewer was the equation of our ionotropic synapse model. This was an unfortunate typing error in our manuscript. The model synapse does in fact correctly implement the synapse as described in Abbott & Marder 1998, with $\tau = (1-s_{\infty}) / k_-$. We thank the reviewer for pointing this out and we will correct this in the text.
>
> To address the reviewers questions:
> - The reviewer asked for clarifications about Figure 1: The photocurrent y-axis is the patch-clamp measured photocurrent measured in pA but normalized by the dark current and the glutamate y-axis is the change in fluorescence of the experimental glutamate indicator from the baseline fluorescence. The green and magenta images show the images shown to the region of interest on the retina in the experiment where the data was collected. The luminance levels of the MNIST squares are converted to photoreceptor stimulus intensities in P*/s, and the model photoreceptors each have an (x, y) location that indicates what pixel of the digit they see. We show only a single horizontal cell in the diagram because this is one of the architectures we train, and showing more horizontal cells would look a bit messy. The BC readout traces are simply example voltage traces of arbitrary size and timescale, depicted only to sketch the task setup. We did indeed not reference this figure in the manuscript and thank the reviewer for pointing this out to us. We will clarify this in the main text of the paper and add references to Fig. 1.
> - Regarding equations, we will add the general form of the gating variable equations to the main text. They differ slightly depending on the ion channel, but their general form is $\frac{dm}{dt}= \alpha(V)(1-m) - \beta(V)m$. We will also add the ion channel equations to the supplementary materials. We also thank the reviewer for pointing out several small formatting errors which we will of course fix in the revised manuscript.
> - Local horizontal cell connections refers to an architecture where each horizontal cell is connected to its 50 nearest cones. This means that cones have the opportunity to influence one another’s activity via horizontal cells, likely being close to what is observed in the real retina. The one-to-one connectivity pattern is designed to model a case in which each cone photoreceptor gets feedback from an electrically isolated segment of horizontal cell. In this architecture, cones still receive horizontal feedback, but they are not able to influence one another via the horizontal cells. We will clarify this further in the text.
> - An intuitive explanation for why horizontal cells could contribute to contrast invariant encodings is that they are positioned to provide photoreceptors with lateral feedback. Lateral feedback is in reference to the case where a strongly excited cell inhibits its neighbors, for example by exciting an inhibitory interneuron. This action can enhance contrast by creating a larger difference between the activity of the two cells. Such inhibition was observed in wet-lab experiments with photoreceptors in the 1950s and 60s; however, other findings since have conflicted our understanding of what horizontal cells do. A potential intuitive explanation why horizontal cells would not contribute to contrast invariant encoding is that highly stimulated photoreceptors themselves perform enough local luminance adaptation so that they do not elicit enough horizontal cell activity to inhibit their neighbors. We are happy to add a summary of these explanations to the abstract.
>
> We agree with the reviewer that more complex and natural stimuli will be an interesting direction for future work. We would like to emphasize, however, that our work is one of the first to train highly-detailed biophysical models on any task such as contrast-invariant classification and our contrast-modulated MNIST provides a reasonable test case for this.
>
> Using more complex stimuli is, however, high on our list of things to study. Because we trained our photoreceptor models on stimuli with color, we have estimated how sensitive each cone is to green and UV light (the two opsins present in mouse cones). Thus, we could also study the role of different cone types in different retinal regions in pattern discrimination. In the current paper, we restrict ourselves to the outer retina’s processing of stimuli with varying contrast and luminance.
>
> To another point of the reviewer, we are happy to provide more discussion of previous models in the discussion section. None of the models in previous work were both biophysically detailed and made predictions about cellular mechanisms responsible for the contrast invariant encoding of visual information. Baek et al. developed a model with similar detail to ours, but its ability to perform any sort of computation was not tested (2020). Another retina simulator developed by Wohrer and Kornprobst used a phenomenological model of the outer retina. This model includes a baked-in mechanism for contrast gain control, but it cannot be used to study what existing biophysical mechanisms might be responsible for such a computation (2009). We can continue to provide detail.
>
> We thank the reviewer for their valuable feedback and kindly ask them to raise their score given our clarifications, additional experiments and modifications to our paper.

---

> > ### Comment · Reviewer_ApBk · 2025-08-04
> > **Reply to rebuttal**
> >
> > Thank you for your detailed and great responses, which address most of my concerns. I will increase my score.

---

> > > ### Author Response · Authors · 2025-08-04
> > >
> > > We thank the reviewer for their positive feedback.

---

> ### Comment · Reviewer_ApBk · 2025-08-05
>
> I have another follow-up question because I saw feedback from other reviewers to the authors. Even though I raised the score, I hope the authors can continue to respond.
> >The one-to-one connectivity pattern is designed to model a case in which each cone photoreceptor gets feedback from an electrically isolated segment of horizontal cell. In this architecture, cones still receive horizontal feedback, but they are not able to influence one another via the horizontal cells.
>
> Does the number of inhibitory neurons impact the response of excitatory neurons to contrast? The network features both lateral inhibition and lateral facilitation. Your rebuttal covers lateral inhibition. I would like to hear from the authors.

---

> > ### Author Response · Authors · 2025-08-05
> >
> > We thank the reviewer for this follow-up question. In our current model, we tested the effect of different photoreceptor-horizontal cell feedback loops in the first experiment (summarized in Table 2).  At some point we also tested some different numbers of locally connected horizontal cells and this did not noticeably affect classification accuracy, also suggesting that the photoreceptor responses were not largely different. We then decided to pursue the contrast experiment in Figure 5 only for one of the networks with 9 locally connected horizontal cells since it was one of the best performing architectures. It would be interesting to perform further experiments testing even more architectures on both of these tasks, but we doubt we can do that until the end of the discussion period considering that these tasks take a while to train. We hope that answers your questions.

---

> > > ### Comment · Reviewer_ApBk · 2025-08-05
> > >
> > > Thanks for the answer. No worries! Our discussion time was extended to the next 48 hours. I think you can try more architectures on both of these tasks, which will greatly enhance your conclusion.

---

> > > > ### Author Response · Authors · 2025-08-05
> > > >
> > > > Yes, we saw the email just now. Unfortunately, in our experience, the models take about 3 days to train. In case we obtain more results until the end of the discussion period, we will let you know. Thanks for the constructive discussion.

---

### Official Review · Reviewer_RpSH · 2025-07-15

**Clarity:** 3
**Significance:** 3
**Originality:** 3
**Rating:** 5
**Confidence:** 4

**Summary:**

This paper introduces a novel framework for training and inspecting models of biological neural circuits, applying it specifically to the under-appreciated processing of the outer retina. The contributions are twofold: first, the authors take the nonlinearities of phototransduction in the outer retina seriously and attempt to understand how they interact with downstream processing (whereas the vast majority of adjacent work make assumptions of linearity). Second, they demonstrate a dual-stage model-fitting procedure that optimizes both biophysical fidelity (using priors and data from literature and experiments) and subsequent performance on an ethologically-plausible task (contrast invariance). Their model, specifically, is a dynamic model of the outer-retina fit to both whole-cell electrophysiology from cones and two-photon glutamate sensor imaging from ribon-synapses, providing a detailed and data-constrained simulation of how light is communicated by the cone photoreceptors to downstream circuitry. They then extend this model to include a variety of architectures that add bipolar and horizontal cells in a manner that enables classification of the MNIST dataset by decoding readout bipolar cell activity. Importantly, they evaluate their model not on MNIST alone, but on out-of-distribution samples that have varied contrast and luminance. The biophyiscal front-end of their model is accurate and captures temporal phenomena (adaptation); the necessity / sufficiency of various aspects of the downstream decoding circuitry are used to make hypotheses about the role of different cell types in the known phenomena of contrast invariant processing.

**Questions:**

1. In figure 5, the comparison of a linear classifier with the biophysical model is a good control, but only one of a few that are needed. In order to test whether it is architectural (not actually due to the learned parameters) the authors should also include the initialized biophysical model, to rule out that the OOD performance is not merely a result of the addition of dynamics. The authors could also evaluate using the same pipeline, with a much simpler dynamical system (see the LNK model in Ozuysal and Baccus, 2014) in order to get a sense of whether the complexity of the dynamics model is actually necessary. Additionally, what if the models are trained on a variety of contrasts, not merely tested on them? One could imagine that varying contrasts could confuse a linear classifier during training, but not your biophysical model.

2. In general, I think the authors are missing an opportunity to investigate the temporal nature of contrast / luminance adaptation. The authors present MNIST data to their biophysical models at varying levels of contrast, and do so in a temporal manner - however, contrast / luminance invariance is a dynamic process that involves the adaptation of a response to the mean contrast or luminance over a specific window of time (Baccus and Meister, 2002). Thus, I would appreciate some exploration of how the model behaves during the transition between high and low contrast regimes, not just evaluated at specific instances of varying contrast. Since contrast invariance is mediated by adaptation, this could be incorporated into the training of the readout network and structure of the task (if the input to the model were, for example, a sequence of frames for one digit with randomly fluctuating luminance and contrast). I imagine a linear model, even trained on inputs with fluctuations in contrast, would have quite a bit of trouble with this task.

3. Given the highly complex biophysical model of cones, I would like to see some preliminary attempt to understand an aspect of the biophysical model of the cones. Since the model is differentiable, the authors could identify which of the parameters / states contributes most to contrast adaptation. Several strategies could be taken - randomly ablating specific states within the dynamics model could identify the crucial mechanisms governing its behavior. Alternatively, looking at the evolution of these state variables during adaptation (or the gradients) could offer clear hypotheses about what aspects of the phototransduction cascade govern contrast invariance.

4. The authors suggest that the highly-complex biophysical model offers an inductive bias that imbues the network with new capabilities (OOD performance on differing contrasts and luminance on MNIST). This is quite a strong claim, given that the machine-learning community already has neural-network primitives that provide similar computational mechanisms without the need for such computationally costly dynamics models (layer-norm, for example). The authors should make explicit that the utility of the model is in gaining insights into their biophysical system - or, if the claim is that some aspect of the biophysical model enables a new computational capability (beyond a vanilla neural network), the authors should draw parallels between more standard machine-learning techniques (layernorm) and provide comparisons.

5. In the retina, contrast adaptation is seen dramatically in amacrine and ganglion cells - but this reviewer's understanding is that it is a much smaller effect in photoreceptors. Please provide some contextualization of the relationship between contrast invariance in the outer retina and the inner retina, and provide a bit more justification for evidence that horizontal cells were ever considered as a putative mechanism thereof.

**Ethical Concerns:**

["NO or VERY MINOR ethics concerns only"]

**Final Justification:**

After thoughtful responses from the authors, I have changed my score to an accept. This is specifically because I think their responses indicate that they have drawn important links to other areas of work and have explored the kinds of controls I was hoping to see in the original work.

**Limitations:**

MNIST is semi-naturalistic but is not really a naturalistic stimulus for any organism. I would encourage the authors to discuss more how their results can and cant be extended to a general understanding of the processing of naturalistic vision with spatiotemporal correlations and structured fluctuations in contrast and luminance. In future work, different tasks could be used that better approximate truly naturalistic vision (cifar, imagenet, video-classification, etc).

**Paper Formatting Concerns:**

No major concerns

**Quality:**

3

**Strengths And Weaknesses:**

Overall, the main strength of this paper is its creative framework for evaluating biophysically-constrained networks on in-silico tasks, as a means of probing the functional impact of specific biophysical parameters. As mentioned by the authors, recent years have brought a large number of papers that optimize models of neural encoding directly to data and by comparing the representations of task-trained networks to experimental data - the author's approach (training to data, then evaluating on an ethologically-meaningful task) has significant potential across a wide range of questions. The paper is written clearly and the approach is well-motivated, however the scientific results of the paper, while well-reasoned, lack important controls and further analyses in order to demonstrate the potential of this new biophysical / task optimized model.

Specifically, the authors motivate their biophysical model architecture by way of its complexity and fidelity to known processing in the outer retina (in addition to the fact that it is differentiable, which this reviewer does not find particularly novel, given the widespread use of autograd functions). It is unclear, however, whether all of this complexity is actually needed for their ultimate conclusions. As described in the paper, their cone-photoreceptor model involves 26 differential equations and this faithfulness is suggested as a mechanism for their downstream task-performance on contrast and luminance invariance testing data. However, the author's do not cite other papers that capture such phenomena using many fewer parameters in a number of different retinal cell-types (including photoreceptors). For example, in "Linking the computational structure of variance adaptation to biophysical mechanisms" (Ozuysal and Baccus, 2014) - contrast adaptation can be captured using a much simpler dynamical system with many fewer parameters (though they do not appear to investigate photoreceptors). However, "Dynamical Adaptation in Photoreceptors" (Clark, ..., Meister, Azaredo da Silveira) do build a model of contrast adaptation in photoreceptors that captures the phenomena. Since much simpler models can capture the phenomena of contrast and luminance invariance - the burden lies with the authors to demonstrate that the complexity of the biophysical model is either useful functionally - or permits some deeper insight. The former could be demonstrated via additional controls. The latter is demonstrated cursorily by suggesting that horizontal cells are not required for contrast invariance.

In regards to the task-evaluation of the biophysical model, the actual performance demonstrated is quite low, given that even simple MLPs can achieve >95% accuracy on MNIST. The reason is presumably that the task is not vanilla MNIST, but the contrast-invariance test. Thus, providing some more standard baselines to show how well standard architectures can perform on OOD evaluation would reassure this reviewer as to the significance of an accuracy of 0.6 (in Table 2). The authors include the "Linear" model, but in order to account for the additional parameters in the biophysical model, the authors could include a biophysical model at initialization or biophysical models with scrambled dynamics parameters.

---

> ### Author Rebuttal · Authors · 2025-07-31
>
> We thank the reviewer for taking the time to review our paper and provide helpful feedback. We are encouraged that the reviewer called our framework “creative,” our model “detailed” and “accurate,” and our paper “well-motivated.”
>
> A main concern of the reviewer is that we do not include more simple controls. For testing out of distribution performance, we have now tested a variety of other linear and nonlinear architectures for comparison. Specifically, we have tested the linear classifier with layer normalization (which performs extremely well out of the training distribution, as expected), a two layer MLP which also performs well out of its training distribution but with a much greater number of trained parameters, the initialized biophysical model as suggested by the reviewer (see answer to Q1 below), and a couple of other references suggested by other reviewers. These experiments have shown us that our biophysical model could be implementing the computations of a linear model with layer normalization, but our biophysical detail allows us to provide an account of how this computation is implemented with the cell’s molecular components.
>
> Similar to these artificial architectures, models such as the one proposed by Clark et al. are phenomenological models, and thus target a different level of understanding of the biophysical system. While phenomenological models can provide interesting insights, they do not provide intermediate steps of computation that can be validated as our model does. To highlight this, we have also plotted the state variables of the phototransduction cascade when performing the classification task on out of distribution stimuli. We are happy to include this as a supplementary figure for our paper, or potentially in the main text if space allows. This figure shows that the photopigment and phosphodiesterase activity change greatly as the contrast moves further out of distribution, but that the cGMP concentration is quite robust, and then in turn the downstream calcium concentration, photocurrent and photoreceptor voltage. Our Fig. 5 currently only shows the glutamate traces as contrast moves further out of distribution, but we are happy to show additional state variables that make predictions about the biophysical mechanisms supporting contrast invariant encoding in even more detail.
>
> The reviewer also voiced concern that our classification accuracy is not high. While reasonable performance is important, the goal of our paper is rather to understand how a biophysically realistic system could implement contrast invariant visual processing and what quantities are most decisive. Biophysical systems are much more constrained than artificial architectures because information is transmitted by a variety of bounded physical quantities.  The biophysical model without the horizontal cells also does not have more trainable parameters for learning the task than the linear classifier, it has 2,000 weights while the linear model has 2,010, and the biophysical model with its initialization parameters only achieves chance accuracy at any contrast level. We have, however, included other reference models as previously mentioned which provide guidance on what computations need to be implemented to solve the task.
>
> Finally, the reviewer notes a limitation that MNIST is not naturalistic. We would like to emphasize, however, that our work is one of the first to train highly-detailed biophysical models on any task such as contrast-invariant classification and our contrast-modulated MNIST provides a reasonable test case for this. We are very excited to see in future work if our results still hold when processing other images and videos.
>
> To address the questions of the reviewer:
> 1. To assure the reviewer that the out of distribution performance is not due to model architecture, we see that the model only achieves chance performance on the task at initialization, even for stimuli within the training distribution. The model weights are initialized from normal distributions centered at values chosen to promote stable dynamics throughout training. To answer the subquestion of task performance if the models are trained on a variety of contrasts: We do train our biophysical models on contrasts chosen randomly in Tab. 2. When training the linear model this way, it actually achieves a high test accuracy of 87%. The linear model is able to achieve high classification accuracy on data within its training distribution.
> 2. The reviewer asks about more analysis on the temporal nature of contrast and luminance adaptation: We think this is a great idea! The linear model we test actually has no concept of time; it simply takes a snapshot at the peak stimulus intensity and uses that for classification, so we would have to reformulate this reference model. However, for this rebuttal, we did briefly test whether our trained biophysical model would maintain its test accuracy on stimuli with contrast fluctuations throughout the stimulation period. The test accuracy in this case only reduced to about 48% (without any additional training). Future work could extend this analysis.
> 3. The reviewer suggests a “preliminary attempt to understand an aspect of the biophysical model of the cones.” We thank the reviewer for this suggestion, and we refer them to our previous discussion of the other state variables in our model, a plot that we can add to the paper.
> 4. We will be more explicit that our goal is to obtain biological insights into contrast-invariant visual encodings while adding our comparison to layer normalization and other architectures as discussed previously.
> 5. A major goal of this paper is to enhance appreciation for the computational power of the outer retina which extends beyond the linear filtering of stimuli. Horizontal cells are positioned to provide lateral feedback to photoreceptors, a phenomenon first observed in the 1950s and 60s in Nobel prize winning work by H. K. Hartline and colleagues. Lateral feedback is in reference to the case where a strongly excited cell inhibits its neighbors, for example by exciting an inhibitory interneuron. This action can enhance contrast by creating a larger difference between the activity of the two cells. Such inhibition was observed in the photoreceptors of Hartline’s horseshoe crab and in other species since; however, other findings have conflicted our understanding of what horizontal cells do. For example, Fred Rieke wrote in 2001 that contrast adaptation was likely to first occur in bipolar cell dendrites after showing that a model of horizontal cell activity did not change at different contrast levels but the bipolar cell model did. The suppression of horizontal cell input to bipolar cells also did not change contrast adaptation in this study. We only briefly discussed this history and motivation in our introduction, and we would be happy to add more information.
>
> We hope that the additional clarifications and experiments increase the confidence of the reviewer in our paper and potentially allows them to raise their score.

---

> > ### Comment · Reviewer_RpSH · 2025-08-06
> >
> > Thank you for the thorough response, and apologies for the delay.
> >
> > Overall, I am mostly satisfied by the responses. The comparison to a linear model with layer-normalization grounds the work more solidly in relation to modern neural-network techniques, and I agree that even though linear+layer norm is a simple operation, the strategy employed is a way to understand *how* the this computation is performed by the biophysical system. Including some more compelling examples of state-variables evolving over time would also highlight this aspect of the work, and so I would strongly suggest doing so.
> >
> > A few remaining questions that I had, to tie my lines of inquiry together:
> > 1. How does the linear+layer-norm model performance compare to the biophysical model?
> > 2. Have you explored models where the parameters of the biophysical system are allowed to be fine-tuned on the MNIST task, in addition to the inner-retina parameters? I am wondering if perhaps you can get the system in the ballpark using the experimentally constrained parameters, but that for full-performance, a fine-tuning step would actually achieve compelling performance without changing the story that emerges from the state variables.
> > 3. Do the state-variable plots you mention tell a story at all, about what parts of the model are responsible for the computation?
> >
> > Overall, if you can include the model controls and explicitly tie the story to layer-norm, show the state-variable trajectories and make some conclusion from them, I am quite satisfied.

---

> > > ### Author Response · Authors · 2025-08-07
> > >
> > > Dear Reviewer, thank you for your questions, and we are indeed happy to show the additional state variables and layer norm results in the paper.
> > > 1. The layer-norm + linear model outperforms the biophysical model in the first task classifying across contrasts with 91% test accuracy and performs similarly to the biophysical model classifying outside of its contrast training distribution (in the sense of the shape of the curve), though its absolute test accuracy is larger at all levels. We attribute the higher test accuracy to the layer-norm + linear model being less constrained by biophysical realism than our model is.
> > > 2. We did not try this approach, but we thank the reviewer for the suggestion! Our model takes too long to train to provide results in the remaining discussion period, but we have future plans to try this.
> > > 3. The state variables plots primarily tell the story that the photoreceptors do a great amount of local adaptation that contributes to the contrast invariant representation. A large part of this local adaptation is achieved by the phototransduction cascade, where the state variable plots show large differences in stimulus resulting in large differences in opsin activity and photodiesterase activity, but smaller differences in downstream cGMP concentration, calcium concentration and photocurrent. We would thus hypothesize that the cGMP concentration dynamics play an important role.

---

### Note · Authors · 2025-08-14

In this work, we introduced a highly detailed model of the outer retina and demonstrated that it can achieve contrast-invariant representations of visual information. We trained thousands of biophysical parameters in multiple model architectures to test the usefulness of various biophysical components including horizontal cells in a variety of connectivity patterns and bipolar cells with realistic ion channels. In general, all of the reviewers considered our work to be novel, and some also found it to be technically impressive (ApBk), creative (RpSH), exceptionally realistic (N9j5), accurate (x6ob, RpSH), and well-motivated (RpSH). The main concerns of the reviewers were the addition of more comparisons to other non-biophysical architectures and more biophysical insights and descriptions of the computations in the outer retina. We addressed these concerns in our rebuttal and the following discussion period. Specifically, we tested different reference models including layer-normalization + logistic classification, a two-layer MLP, and various thresholding techniques + logistic classification. We also provided additional analysis of our biophysical model’s state variables and explanations surrounding the outer retina’s ability to perform contrast-invariant encoding. We thank the reviewers for their valuable feedback and for actively engaging in the discussion period which has further strengthened our paper. This work demonstrates how machine learning can advance our understanding of retinal computations, and we would look forward to presenting it at the NeurIPS conference.

---

### Decision · Program_Chairs · 2025-09-17

**Decision:**

Accept (poster)

**Comment:**

This paper proposes a biophysically plausible model of the outer retina adapted to gradient-based learning with modern machine learning tools. While previous models of several components used here exist, along with some similar efforts, this model is unique in combining realistic biophysics with tunable nonlinearities in a fully differentiable framework that can be used to investigate early-stage visual processing. The authors perform a number of experiments in this framework to delineate the contributions of various circuit elements, finding that inhibitory feedback through horizontal cells provided only limited improvement in the tasks they consider.

Reviewers applauded the careful nature of the modeling, which gathers together a good deal of previous literature in a modern package that can be optimized using JAX. They were likewise positively impressed with the potential for understanding processing in neural circuits. Weaknesses centered on the performance evaluation and baselines, which included fairly poor performance on MNIST and comparison only with a linear model.

Discussion primarily focused on two points:
1. **Is MNIST, in which gray pixels play a minimally informative role, a good task on which to judge contrast-invariant processing in the model?** That is, does contrast significantly modulate task difficulty in a way that illustrates the model's ability to compensate. Here, the authors acknowledged that a simple binarization threshold is sufficient to remove effects of contrast, though they argued that adaptively ignoring contrast changes requires such a threshold to be set, and this is something analogous to what the model does.

2. **The performance of the model on MNIST is well below that of even simple neural networks, and only a linear model is used as a baseline comparison.** The authors responded by training additional simple baselines (including logistic classification on top of a layer norm), though they argued that performance metrics _per se_ were not the point of the modeling.

While reviewers were appreciative of the second point, objections remained to the first. In a task where contrast is not directly relevant to performance (or can be easily compensated for), it is debatable whether any level of performance assesses contrast (in)sensitivity. Authors would likely do better to choose a separate task in which contrast more clearly modulates problem difficulty.

On balance, however, the careful nature of the modeling and its potential utility in neuroscience research argue for acceptance of the paper.